

# A comprehensive review of gait analysis using deep learning approaches in criminal investigation

Sai Thu Ya Aung and  Worapan Kusakunniran

Faculty of Information and Communication Technology, Mahidol University, Salaya, Nakhon Pathom, Thailand

## ABSTRACT

Despite the growing worries expressed by privacy supporters about the extensive adoption of gait biometrics, research in this field has been moving forward swiftly. Deep learning, a powerful technology that enables computers to learn from data, has found its way into criminal investigations involving gait. In this survey, the literature of gait analysis concerning criminal investigation is discussed with a comprehensive overview of developments in gait analysis with deep neural networks. Firstly, terminologies and factors regarding human gait with scenarios related to crime are discussed. Subsequently, the areas and domains corresponding to criminal investigation that can be tackled by gait analysis are discussed. Also, deep learning methods for gait analysis and how they can be applied in criminal investigations are presented. Then, gait analysis techniques and approaches using deep learning methods including currently available datasets are mentioned. Moreover, crime-related video datasets are presented with literature on deep learning-based anomaly detection with gait human poses. Finally, challenges regarding gait analysis in criminal investigations are presented with open research issues.

Corresponding author
Worapan Kusakunniran,
worapan.kun@mahidol.edu

# INTRODUCTION

Gait is a pattern of walking and is considered a common trait of humans. However, in terms of analysis, it is considered one of the most complicated and sophisticated approaches. Human gait has been examined and researched since a long time ago and is regarded as unique and indicates individuality in humans. Gait is characteristic of one's individuality; it is considered to be an accepted form of unobtrusive biometric (*Jain, Bolle & Pankanti, 1996*). With the arrival of surveillance technology, identification and verification of human individuals and their behaviors in surveillance videos using gait biometrics has become a widely discussed topic as human gait can be captured from a distance. Another potential of utilizing gait biometrics is its non-invasiveness nature that does not require physical contact or the use of invasive devices with the individuals being analyzed eliminating the need for individuals to cooperate or engage. This characteristic of being non-invasive makes gait biometric and appropriate for identification particularly suitable for situations where direct interaction with human subjects in crime-related activities is not feasible.

Biometric identification (*Boulgouris, Hatzinakos & Plataniotis, 2005*) has become a powerful tool in modern criminal investigations, aiding law enforcement agencies in identifying and apprehending suspects. Computer vision and machine learning techniques leverage gait analysis in criminal investigation to capture and analyze a person's walking pattern from surveillance videos or other sources. By extracting biometrics features from the gait, such as step length, stride duration, and several human postures which are extracted with sophisticated machine learning algorithms and can create a unique gait signature or profile of each individual. The extracted features are then saved in data silos for security and forensic purposes. This gait signature or features of the suspected person from a crime scene video can then be compared against databases to identify potential perpetrators or to corroborate the identity of known individuals captured in surveillance footage. Unlike other biometric modalities, gait analysis does not require explicit cooperation of the subjects, and analysis can be performed remotely.

As a rapid growth in the number of security cameras, crime scenes are recorded in surveillance systems, however, perpetrators involved in crime scenes cannot be identified easily using face recognition as faces can be concealed easily and intentionally in some crime cases like robberies. Many recent research studies have found that gait recognition holds greater suitability for forensic purposes due to the fact that other identifying characteristics often associated with the crime scene can be obscured or concealed and gait motion is the only feature captured in the crime scene videos from surveillance cameras (*Macoveciuc, Rando & Borrion, 2019*).

This comprehensive review of gait analysis aims to explore the various aspects of gait analysis for criminal investigation, including existing techniques, deep learning methods, available datasets, and challenges. Firstly, the rationale and aim of this study are pointed out along with the intended audience. The fundamental idea of human gait is mentioned in the "Human Gait" section with essential taxonomies, along with gait in crime-related scenarios. The impacted areas and domains related to gait analysis in criminal investigation are presented with seven subsections. Furthermore, deep learning architectures are discussed. In the same way, deep learning-based gait analysis is mentioned emphasizing gait recognition and pose estimation techniques. Recent approaches to gait analysis on crime-related scenarios using deep learning are then discussed. Moreover, challenges and future aspects regarding gait analysis in criminal investigations are summarized.

## RATIONALE AND INTENDED AUDIENCE

The use of gait analysis in criminal investigations has gained significant attention in recent years, particularly with the advent of deep learning approaches (*Wang, 2020*). In the context of criminal investigations, gait analysis can be a valuable tool for identifying suspects, especially when other biometric identifiers such as facial recognition or fingerprints are not available (*Bouchrika et al., 2011b*). Moreover, gait analysis has several advantages over other biometric modalities. For example, gait patterns are more difficult to disguise or manipulate than facial features or fingerprints, making gait analysis a more robust method for identification. Additionally, gait analysis can be performed at a distance, without the

need for physical contact or close proximity to the individual, which can be beneficial in surveillance and forensic applications (*Tan, Huang & Yu, 2012*). Deep learning approaches can be used to analyze large datasets of gait patterns, making it possible to identify individuals even when the quality of the video footage is poor (*Li, Zhang & Li, 2020*) and can be used to analyze gait patterns and identify individuals with a high degree of accuracy.

However, despite the potential benefits of gait analysis, several challenges and limitations need to be addressed. For instance, gait patterns can be affected by various factors such as clothing, footwear, and walking surface, which can impact the accuracy of gait analysis. Moreover, the use of gait analysis in criminal investigations raises several ethical and legal concerns, such as the potential for misidentification and the impact on individual privacy (*Liu, Zhang & Li, 2018*; *Jain, Ross & Prabhakar, 2011*). A comprehensive review of gait analysis using deep learning approaches in criminal investigation is therefore necessary to explore the potential of this technology in criminal investigations and to identify areas for future research and development. This review aims to provide a thorough examination of the current state of gait analysis in deep learning and criminal investigation, including the advantages and limitations of deep learning approaches, and the challenges and limitations of gait analysis. By synthesizing the existing literature and identifying gaps in current research, this review aims to provide a foundation for the development of more accurate, robust, and reliable gait analysis systems for use in criminal investigations.

This article is intended for researchers, law enforcement personnel, and forensic experts who are interested in the current state of gait analysis using deep learning for criminal investigations, mentioning potential benefits and challenges of using deep learning techniques, and the available datasets for gait recognition and criminal actions detection. This paper aims to offer a thorough examination of the intersection between deep learning and gait analysis in the context of criminal investigations, providing a comprehensive review of the literature, and highlighting the gaps and opportunities for further research. It not only outlines the advantages and limitations of current approaches but also identifies key challenges, ethical considerations, and areas where current research is lacking. In this literature review, the contributions that this paper aims to address are:

- Highlighting the potential of deep learning techniques in improving the accuracy and robustness of gait analysis systems.
- Identifying and discussing the practical challenges and limitations of implementing gait analysis in criminal investigations.
- Identifying gaps and opportunities for further research that can lead to the development of more accurate and reliable gait analysis approaches.

## REVIEW METHODOLOGY

The review approach applied in this survey consists of searching information and articles from general to specific. We started with the general concept of human gait, basic concepts of gait, and appliance of gait in criminal investigation. Then deep learning architectures and deep learning-based gait analysis approaches are reviewed. Furthermore, articles mainly

focused on gait analysis and pose estimation using deep learning methods that can be applied in the domains of criminal investigations are explored with practical applications.

To ensure a comprehensive review, we applied keyword-based search in peer-reviewed publications databases of IEEE Xplore, Scopus, Science Direct, ACM Digital Library, and Google Scholar. We focus on articles that were published in 2018 and beyond (until 2023) to include the latest state-of-the-art findings. For keyword searches, we use specific keywords for each relevant area and combine them. The search query is prepared as ((("Gait" OR "Pose" OR "Pose Estimation") AND ("Deep Learning" OR "Machine Learning" OR "CNN" OR "RNN" OR "GRU" OR "LSTM" OR "Autoencoder" OR "GAN") OR ("Crime" OR "Criminal" OR "Forensic")) and searched on the publication databases.

From the selection of articles extracted with the search query, inclusion, and exclusion criteria are set as follows:

Inclusion criteria:

- Peer-reviewed articles and conference articles
- Articles from the year 2018 to 2023
- Articles written in English-language
- Studies emphasizing human gait and deep learning approaches or criminal investigation

Exclusion criteria:

- Non-peer-reviewed articles, such as preprints and theses
- Articles not written in English
- Articles before the year 2018
- Duplicate articles.

From the initial search with the query in each publication database, we found a total of 184 articles, and 170 articles were left out after applying a date filter to include from the year 2018 to 2023. After applying inclusion and exclusion criteria, the final 79 articles are used in this review. The abstract and content of selected articles are reviewed in detail to investigate their methods, datasets, and possible research gaps. The process of this review methodology is depicted in Fig. 1.

## HUMAN GAIT

Gait cycle (*Whittle, 2014*) is a series of cyclic events during walking involving complex interactions between muscles, bones, nerves, and joints. Despite its sophisticated nature, gait serves as an expressive marker by providing insights into various aspects and has been studied for centuries. Research related to gait analysis involves human walking, and exploring the mechanisms of human lower extremities and factors involved governing the functionality.

### Gait cycle

During the gait cycle, several events occur and primarily two phases are involved which are the stance phase and the swing phase. Generally, the stance phase is the phase when the reference foot touches the ground the whole time which means it begins when the reference

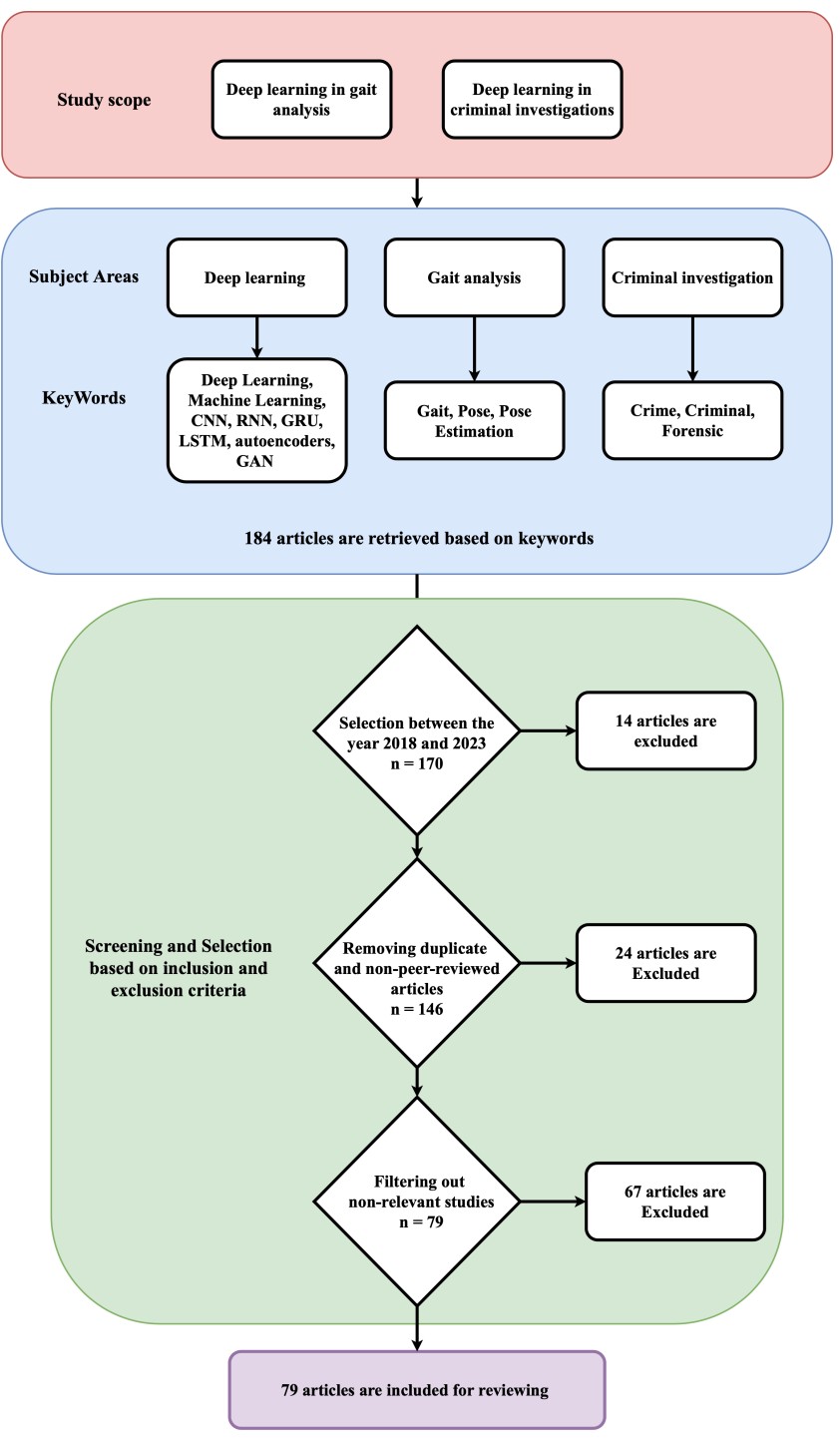

**Figure 1  Review methodology.**

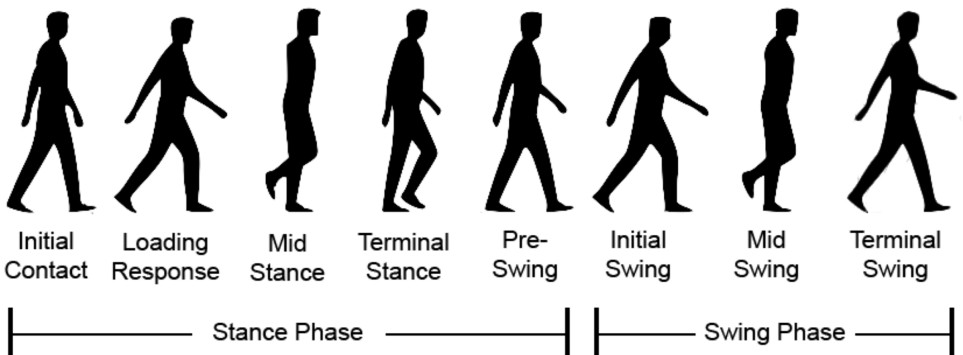

**Figure 2** Events and phases happened during a gait cycle.

foot with the heel contact till the same foot makes the toe off which approximately takes 60% of the gait cycle. During the stance phase, initial contact, loading response, mid-stance, terminal stance, and pre-swing events occur. The duration when the reference is off the ground and swinging is known as the swing phase and it occupies 40% of the gait cycle and initial swing, mid swing, and terminal swing events are involved (*Kharb et al., 2011*). The gait cycle, gait phases, and events are shown in Fig. 2.

Initial contact is also known as heel strike and it is the initial event of the stance phase which happens when the heel of the reference foot touches the ground and it is the beginning of the loading response. Loading response starts when the reference foot makes the initial contact with the ground to give support for the other foot to lift for swing. Mid stance is followed after loading response which begins with the opposite reference foot lifting off and the body weight is directly over the reference foot and the opposite reference foot acts as a single support. The next event is the terminal stance which involves hip extension to maintain single support until before pre-swing as shown in Fig. 2. It happens till the weight of the body is over the reference foot and ends when the opposite reference foot touches the ground. The stance phase with pre-swing is considered to begin when the opposite reference foot touches the ground and ends when the reference foot makes a toe-off.

Initial swing is the initial event of the swing phase and in this event, the reference foot is in flexion mode to be pushed and lifted off the ground till mid-swing. During mid-swing, the weight of the body is over the opposite reference foot and the thigh of the reference foot is at its peak for swing. Terminal swing is the final event of the swing phase and gait cycle and ends when the reference foot touches the ground to start the next gait cycle.

### Spatial and temporal gait parameters

Gait parameters provide a comprehensive understanding of the gait pattern of individuals. The simplest gait parameters are the distance or spatial parameters, such as step and stride length, step width, and cadence. These parameters reflect the distance covered during walking and are useful in identifying changes in gait patterns.

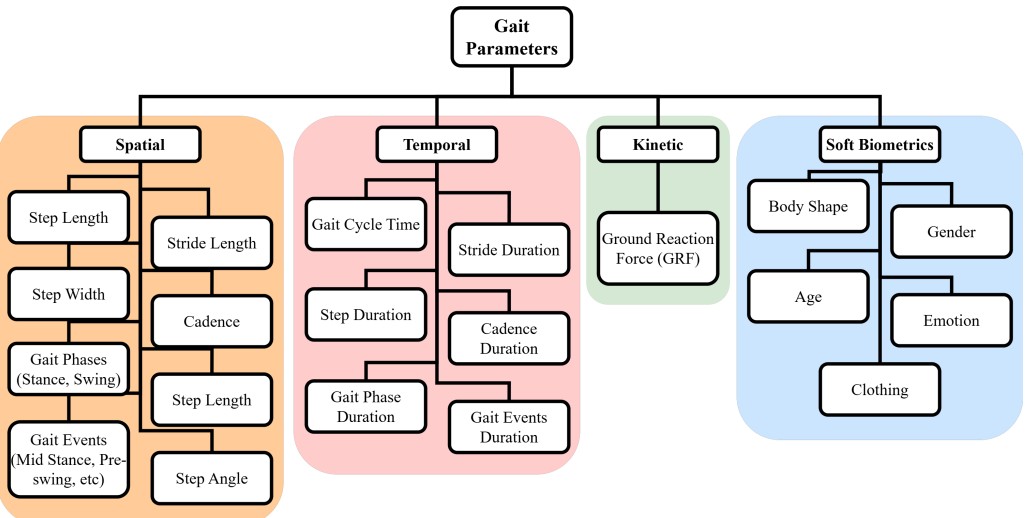

**Figure 3  Gait parameters: comprehensive tree diagram.**

Stride length is defined as the distance covered from the initial contact of the reference foot to the ground till it makes another contact that involves two steps and captures a full cycle of a single leg's movement. Step length refers to the distance between the initial contact of one foot with the ground and the initial contact of the opposite foot with the ground. Step width, on the other hand, is the distance measured between the line of progression of the left foot and the line of progression of the right foot. Spatial parameters in gait are used to measure gait asymmetries related to the distance between steps and stride lengths. Cadence is the total number of steps that happen per unit of time, usually measured in steps per minute.

Temporal parameters in gait describe the time duration between different events in the gait cycle. The simplest temporal parameter is the gait cycle time, which is the time from one heel strike to the next heel strike of the same foot. Other temporal gait parameters involve step and stride duration and cadence. It is typically measured in seconds and is used to calculate other temporal parameters such as the stance phase, swing phase, and double support phase (*Grabiner, Biswas & Grabiner, 2001*). The parameters used in human gait are shown in Fig. 3.

### Kinetic parameters

Kinetic parameters in gait refer to the analysis of forces acting on the body during walking. The measurement of these parameters is important for evaluating gait and assessing the risk of injury. The most commonly used kinetic parameter in gait analysis is ground reaction force (GRF) (*Oh, Choi & Mun, 2013*).

GRFs are the forces that the ground exerts on the body during walking. They are measured using force plates, which are embedded in the floor. GRFs are commonly used to calculate the center of pressure (COP) trajectory, which is the point of application of the net force acting on the foot during the stance phase. COP is an important parameter for

understanding balance and stability during walking. With kinetic parameters, inefficient movement patterns can be identified and used for quantifying abnormalities in gait.

### Soft biometrics parameters in gait

Soft biometrics parameters in gait are characteristics that can be used to infer the identity or demographic information of an individual based on their gait pattern (*Nixon et al., 2015*). Unlike traditional biometrics such as fingerprint or facial recognition, soft biometrics parameters are more subjective and less reliable, but can still provide useful information for various applications such as surveillance and forensic investigations. Some examples of soft biometrics parameters in gait include:

Body shape: The shape of an individual's body can affect their gait patterns, such as their stride length and step width, and provide information for further identification.

Gender: There are differences in gait patterns between males and females, such as the degree of pelvic rotation and arm swing. For instance, males have higher gait speed and stride length than females which can be applied for gender confirmation.

Age: As people age, their gait pattern changes due to factors such as decreased muscle strength and joint mobility that lead to walking slower and shorter step lengths and can be identified age from their gait pattern.

Emotion: Emotional states can impact gait as the inner feelings of an individual have a connection to gait patterns.

Clothing: The type of clothing an individual wears can affect their gait pattern, such as tightness and weight of the clothing. Clothing affects gait in several ways and is useful in gait analysis and identification.

Shoes: Shoes are a significant gait parameter as they can significantly influence gait impacting comfort and stability, such as the height of the heel and the level of cushioning.

Overall, soft biometrics parameters in gait can provide additional information to traditional biometrics and be used as additional parameters for gait analysis.

## Inhabiting factors in gait

While gait analysis can provide useful information for various fields, there are some inhibiting factors that can affect the accuracy of the analysis. One of the main inhibiting factors is the presence of an abnormal gait, which can be caused by injuries, disabilities, or other medical conditions. Abnormal gait patterns can make it difficult to identify certain gait parameters, leading to inaccurate results. Another factor is the environment in which the analysis is conducted. Other factors that need to be considered are

| | |
|---|---|
| Clothing: | Clothing can obstruct or cover certain parts of the body, which can make it difficult to capture the required data. For example, loose-fitting clothing may hide body contours, making it hard to accurately capture limb movements. |
| Environment: | The environment in which gait analysis takes place can affect the accuracy of the results. For example, uneven terrain, wet surfaces, or obstacles may alter an individual's gait and introduce measurement errors. |
| Medical conditions: | Certain medical conditions such as arthritis, hip or knee replacements, or amputations can alter an individual's gait, which can make it difficult to determine normal or abnormal gait patterns. |

Pain:       Pain can alter an individual's gait, resulting in an abnormal walking pattern. For instance, a person with a painful knee may adopt a limp, which can make it difficult to determine their normal gait.

Fatigue:       Fatigue can also affect an individual's gait. If the person is tired or has been walking for an extended period of time, they may adopt an abnormal gait pattern, making it difficult to obtain accurate gait analysis data.

Age:       As people age, they may experience a decline in physical abilities, such as muscle strength and flexibility, which can alter their gait. This may make it difficult to compare their gait to younger individuals or to determine what is normal or abnormal for their age group.

These inhibiting factors can impact the accuracy and reliability of gait analysis and therefore should be taken into account when interpreting gait analysis results.

## Normal and abnormal gait

Gait analysis has been used in criminal investigations for decades and can provide valuable information about suspects in cases where other forms of identification are unavailable. Normal gait refers to how a person walks under typical, emotionally, and physically healthy conditions. Characteristics of normal gait can include a steady rhythm, even stride length and width, and a smooth heel-to-toe motion. On the other hand, abnormal gait refers to any deviations from the expected normal gait pattern. These deviations may be due to a variety of factors, such as injuries, neurological disorders, physical disabilities, intoxication, or emotional distress.

Disturbance behavior, such as aggressive or threatening behavior from the influence of drugs or alcohol, and in such cases, abnormal gait can be an indicator. Scissoring gait is one of the abnormal gaits where legs are moving in a scissor-like manner, and the individual can be seen in certain intoxication. Ataxic gait is usually seen in an alcohol-intoxicated person who shows unsteady gait with staggering or stumbling. With proper gait analysis techniques, abnormal gait can be detected, and security personnel can be informed to take appropriate action to ensure safety.

In criminal investigations, abnormal gait patterns can be used as a potential means of identification. Suppose surveillance footage or eyewitness accounts describe a suspect with a noticeable limp or other abnormal gait pattern. In that case, this information can be used to narrow down the pool of potential suspects. Additionally, gait analysis can be used to compare a suspect's gait to that of a person seen in surveillance footage or other evidence, potentially providing further evidence for or against the suspect's involvement in a crime.

There are several scenarios of criminal investigation based on abnormal gait patterns. In 2004, there was an incident of a bank robbery that was handled and investigated by the Department of Forensic Anthropology at the University of Copenhagen. When the police observed the video footage of the crime scene, they found out that the perpetrator had an abnormal gait pattern. For this reason, the police consulted with gait analysis experts to help with the investigation. Gait analysis experts requested another recording of the suspect walking at the same angle for comparison with the crime footage. Finally, gait analysis divulged that there are matches between the suspect and the perpetrator as

the forward rotated feed and inverted left ankle during the stance phase. Other posture resemblances, including a restless stance and interior head positioning, are found using photogrammetry. Some incongruities were found during the gait analysis, such as a wider stance and the trunk slightly leaning forward with an elevated shoulder. These observations were presented in court, and the culprit was convicted of robbery while the court pointed out that gait analysis is a valuable tool (*Lynnerup & Vedel, 2005*).

In another similar case that happened in the U.K., a burglar was apprehended by a security force because of his distinctive walking way and abnormal gait. The police officers and a podiatrist assessed videos of the perpetrator and suspects. Upon analyzing and performing a posture assessment, they could pinpoint one of the suspects by a significant similarity in gait between the perpetrator and the suspect. The gait-based analysis was then submitted as part of the evidence, and the perpetrator was prosecuted (*Bouchrika et al., 2011a*).

In both of these cases, the suspects' abnormal gait patterns served as critical identifying factors by offering unique insights and crucial evidence for law enforcement. The analysis and recognition of irregularities in a person's gait are becoming significant identifiers that complement traditional identification. Moreover, abnormal gait analysis becomes valuable when other identification traits like face and fingerprint are unavailable or manipulated.

## Gait in crime scene videos

Gait analysis is an essential tool in forensic investigations, particularly in the identification of individuals captured in surveillance videos. By analyzing an individual's gait, investigators can identify and compare specific characteristics such as step length, stride length, and cadence. These parameters can provide critical evidence in criminal cases, including identifying suspects and linking them to a specific crime scene.

Gait analysis in crime scene videos involves capturing and analyzing an individual's gait pattern as they move through a particular area. Investigators can use this information to determine the individual's height, weight, and other physical characteristics. This can help narrow down a pool of suspects and lead to a more targeted investigation. Gait analysis is particularly useful in cases where other identifying features, such as a suspect's face, are obscured or not visible in the video footage. One of the challenges of gait analysis in crime scene videos is the quality of the video footage. Low-quality videos or videos captured from a distance may not provide sufficient detail for accurate analysis. Additionally, the presence of obstructions, such as furniture or other people in the video, can make it difficult to track an individual's gait accurately. The lack of consistent lighting or changes in lighting conditions can also make it challenging to analyze an individual's gait accurately.

Another factor that can affect gait analysis is the individual's clothing or footwear. Certain types of shoes or clothing can alter an individual's gait pattern, making it more difficult to identify them accurately. Environmental factors such as weather conditions, the surface on which the individual is walking, and the slope of the ground can also impact an individual's gait pattern. Despite these challenges, gait analysis remains a powerful tool in forensic investigations. By identifying unique characteristics in an individual's gait pattern, investigators can provide evidence linking suspects to specific crimes. As

technology continues to advance, gait analysis will likely become even more accurate and widely used in criminal investigations.

In 2017, a series of robberies occurred in a city in China, and the suspect managed to evade capture by frequently changing his appearance, wearing different clothes, and using various escape routes. Traditional methods of identification, such as facial recognition, were not effective in tracking him down. Law enforcement agencies turned to gait analysis as a potential solution to identify the suspect. They collected surveillance footage from multiple crime scenes, focusing on the suspect's walking patterns during the robberies. Using specialized software and algorithms for gait recognition, forensic experts analyzed the suspect's gait from the video footage. Gait recognition involves extracting specific features from an individual's walking style and comparing them to a database of known gait patterns. After comparing the suspect's gait patterns with the database, the experts found a potential match with an individual from a previous unrelated case. The suspect from the previous case had a distinct and unique gait that resembled the gait of the robber in the current series of crimes. With this lead, the investigators looked deeper into the suspect's background and criminal history. They discovered that the individual previously identified through gait analysis had a history of robberies and other criminal activities. Based on this breakthrough, law enforcement agencies intensified their search efforts, and officers on the ground focused on areas where the previous suspect was known to frequent. Eventually, they apprehended the suspect, and upon further investigation, he was found to be responsible for the recent series of robberies as well (*Dai, 2019*).

This real scenario demonstrates the practical application of gait analysis in criminal investigations. Gait analysis can be a valuable tool when other traditional methods of identification, such as facial recognition, are insufficient or inconclusive. By leveraging unique physical traits, such as gait patterns, law enforcement can make significant breakthroughs in identifying and apprehending suspects involved in criminal activities. However, it's essential to use gait analysis in conjunction with other evidence and investigative techniques to ensure accurate identification and a successful resolution of the case.

## IMPACTED AREAS RELATED TO CRIMINAL INVESTIGATION

The integration of deep learning algorithms with video surveillance systems has opened up new avenues for enhancing security measures and improving forensic analysis in criminal investigation. This shows applications and adaptations of deep learning in surveillance and criminal investigation, with a particular focus on violence behavior detection, crowd control, and emotion detection in gait analysis. The incorporation of deep learning models has significantly boosted the accuracy and efficiency of surveillance cameras in identifying and tracking suspicious activities. By enabling real-time video analysis, deep learning-powered surveillance systems can promptly detect potential threats, thus empowering law enforcement agencies to take preemptive measures and ensure public safety.

In a criminal investigation, deep learning techniques have proven invaluable in analyzing large volumes of visual data, including images and videos, to extract crucial evidence.

Forensic analysis, which is often a time-consuming and labor-intensive process, has witnessed tremendous acceleration through the use of deep learning algorithms. These algorithms aid in fingerprint recognition, facial identification, and object detection, processing the identification and apprehension of criminals while supporting the reliability of forensic evidence presented in courtrooms.

## Surveillance

The valuable insights and evidence gained from the surveillance camera can be used in crime investigation, especially with gait recognition. The video footage of individuals in public spaces captured by the surveillance camera allows the identification and tracking of each individual movement as normal or suspicious. With the help of gait analysis, valuable information can be extracted to identify the specific person in a situation when their faces may not be obscured or invisible. In such cases, instead of using facial recognition, gait recognition is another available option to identify individuals.

Moreover, surveillance cameras in public areas, for instance, streets, malls, or transportation stations, can catch important footage of criminal activities. Combining gait analysis with this footage can leverage to identify main suspects based on their distinctive walking structure or patterns during short period of time. The extracted gait characteristics from the crime scene video can also be used to compare with known suspects of databases, the process of crime investigation can narrow down their search and potentially link suspects of the crime.

In the aspect of forensic investigations, video from surveillance can give crucial evidence and clues for reconstructing the events and establishing timelines. The movement patterns of criminals which are extracted from gait analysis can aid investigators in understanding their activities, locations, and possible accomplices. By monitoring human gait and abnormal behavior from these surveillance cameras, criminal activities or suspicious behaviors can be detected in real time.

One of the scenarios is security professionals may be able to prevent crime or swiftly respond to security risks by using gait analysis to help them spot suspicious behavior or people with unusual stride patterns. Accident or criminal investigations can benefit greatly from the evidence that surveillance cameras are placed at crossings, on highways, or in public transportation vehicles. Investigators can ascertain the movements and behaviors of those involved in accidents by analyzing gait patterns, which helps to establish liability and contributes to a thorough comprehension of the situation.

Systems for personal identification and access control can also incorporate gait analysis for entrance surveillance. This may be helpful in limited regions, secure facilities, or even to confirm people's identities when more conventional identification methods might not be practical or trustworthy.

Gait analysis is greatly aided by surveillance cameras and video footage, which provide insightful information and evidence for identifying people, supporting criminal investigations, assisting forensic analysis, enhancing security, assisting accident investigations, and facilitating personal identification and access control systems. They help

with many investigative and security applications by adding another layer of information when combined with gait analysis tools.

## Crime investigation

Due to the consecutive wave of technology, gait analysis plays a crucial role in investigating criminal activities from the video footage of surveillance cameras. Gait analysis is a special technique that focuses on analyzing the unique walking structure and patterns of individuals. Therefore, gait analysis aids crime investigation in the identification and tracking of culprits.

Nowadays, surveillance cameras are installed in crowded areas to enhance and maintain public safety. These cameras may be fixed or rotatable to provide coverage of specific locations and continuously record video footage, capturing the movements of individuals in the monitored areas. Therefore, these records are the main source of data for gait analysis. Since gait analysis is the study of an individual's walking pattern, it is based on the understanding that each person has a unique gait signature that can be used for identification purposes.

When a crime is being investigated, specific people of interest who may be involved in illegal activity are found after carefully reviewing the surveillance camera footage. These people are subsequently subjected to the gait analysis procedure to obtain their gait traits. Using computer vision and image processing techniques, gait elements from the video data are retrieved, including joint angles, foot placement, and body movements. Each person's gait profile is created using these characteristics. The gait profiles of suspects are compared to established databases or other reference samples to find probable matches. The gait features may be compared manually or automatically, utilizing algorithms for recognition.

After potential matches are found, extra research is done to collect proof connecting the suspect to the crime. To do this, new surveillance footage, witness accounts, or other forensic evidence may be compared to the suspect's gait profile (*Larsen, Simonsen & Lynnerup, 2008*). Gait analysis in criminal investigations frequently necessitates the assistance of forensic experts, who are trained to examine gait patterns and testify in court. They can considerably advance the investigation with their analysis and interpretation of the gait evidence.

It is crucial that gait analysis from surveillance camera footage is only one component of the whole investigation. It is frequently integrated with additional evidence, including witness testimony, DNA analysis, facial recognition, and conventional investigation techniques, to create a thorough case against a suspect.

Investigators can better identify and monitor suspects, connect people to specific illegal actions, and produce more evidence for criminal investigations by using gait analysis from security camera video footage. In situations where conventional identification methods can be insufficient or unavailable, this methodology has shown to be useful (*Amboni, Barone & Hausdorff, 2013*).

## Forensic analysis

Gait analysis is a useful tool in forensic analysis for gathering and examining forensic data. The study of a person's gait and movement characteristics is called gait analysis, often

known as forensic gait analysis or forensic biomechanics (*Larsen, Simonsen & Lynnerup, 2008*). It can offer critical information and assistance in a variety of forensic investigations.

Gait analysis can be used to identify people based on their distinctive walking styles. When more conventional identification techniques, like fingerprints or DNA evidence, are unavailable or inconclusive, this becomes more helpful. In order to establish a person's presence at the site, forensic professionals can match the gait patterns seen in surveillance footage or footage from the crime scene with those of known people or suspects.

Analyzing a person's gait can help connect suspects to illegal activity. Forensic professionals can match up gait patterns observed in surveillance footage or footage from a crime scene with the gait traits of known suspects. This research can show a possible connection between a suspect's gait and the crime, supplying crucial information for investigators.

Establishing timings and the motions of people during a crime or event can be helped by gait analysis. Investigators can ascertain the chronology of events, the movements of suspects or victims, and their interactions inside a crime scene by analyzing the stride patterns collected in surveillance videos. This information helps to reconstruct the crime and provides a more thorough knowledge of the situation.

Analyzing a person's gait can help determine how reliable a witness is. Gait analysis can be used to evaluate the accuracy of statements made by eyewitnesses about the physical traits or movements of suspects. Forensic professionals can assess the validity of eyewitness statements by contrasting the described gait with the actual gait patterns recorded in films. How trustworthy a witness is can be ascertained by looking at their gait. Eyewitness descriptions of suspects' physical characteristics or movements can be verified or refuted using gait analysis. By comparing the described gait with the actual gait patterns captured on film, forensic experts can evaluate the reliability of eyewitness accounts.

Gait analysis is a useful tool in forensic medicine for establishing time frames, pinpointing suspects, and gathering expert testimony. It aids in the investigation of crime scenes and the pursuit of justice by fusing the scientific analysis of gait patterns with other forensic data.

## Violent behavior detection

Gait analysis techniques are used in the field of behavioral analysis and security for the detection of violent conduct. It seeks to recognize and detect aggressive or violent conduct based on a person's gait and movement characteristics. Gait analysis makes it easier to spot potential aggressive or violent intent by looking at gait factors such as stride length, pace, posture, and body motions (*Saleh & Tahir, 2020*).

Analysis of a person's gait can uncover behavioral clues linked to aggressive behavior. These warning signs include an aggressive body posture, abnormal motions, odd gait patterns, quick or brisk steps, or an aggressive temperament in general. Security systems can identify people who display possible violent tendencies by looking at certain behavioral cues. To recognize and flag people demonstrating potentially aggressive conduct, violence behavior detection employing gait analysis is frequently integrated into surveillance systems, such as video monitoring systems. Video footage of public areas, entrances, or locations

where heightened security is a concern is captured by surveillance cameras. To find any unusual or aggressive movement patterns, the gait analysis algorithms can examine this video in real-time or after post-processing.

Violence behavior detection systems use cutting-edge machine learning algorithms and pattern recognition approaches. To learn and recognize patterns suggestive of hostility or violence, these algorithms are trained on extensive datasets of gait patterns and associated behaviors. The technology may recognize and issue alarms when it identifies potentially aggressive behavior by continuously evaluating real-time video feeds or recorded data. Gait analysis is used to detect violent behavior in order to provide early warning and prevent violent situations. Security staff can identify disputes or step in before they get out of hand by identifying and alerting those who may act violently. Maintaining public safety may include notifying security staff, setting off alarms, or putting in place the necessary security measures.

Gait analysis-based systems for detecting violent behavior can be combined with other security measures like access control or security checkpoints. Gait analysis is incorporated as an additional layer of protection to improve the overall effectiveness of security measures and aid in the early detection of potential threats. It is significant to emphasize that the use of gait analysis to detect violent behavior involves ethical questions about privacy and potential biases (*Demiris et al., 2008*). Implementing such systems with appropriate consent, transparency, and adherence to privacy laws is essential. To reduce false positives and guarantee impartial findings, ongoing monitoring and review of the system's fairness and accuracy is also required. Violence behavior detection using human gait and poses has become an interesting topic as it helps to avoid violent situations and maintain public safety by fusing behavioral analysis, technology improvements, and surveillance technologies.

## Public safety

Using gait analysis technology for public safety requires finding and following people who might be a threat to the safety of others. This system can automatically detect and identify suspicious people based on their walking patterns in public places like airports, train stations, and malls. With the aid of gait analysis, atypical walking patterns, such as limping or shuffling, that may be a sign of an injury or impairment can be found. Additionally, it can be used to spot people who are moving too quickly, too slowly, or in a certain direction, all of which could be signs of unethical behavior.

Gait analysis for public safety is still a relatively new technology, but it has the potential to be a useful tool for security and law enforcement organizations in preserving public safety and reducing crime. When using this technology, it is crucial to take privacy and civil rights concerns into account.

By using gait analysis tools, public safety can be improved. Gait analysis can assist in locating those exhibiting suspicious or unusual behavior. Security systems can flag and notify authorities about people exhibiting potentially dangerous or suspicious gait characteristics by evaluating a person's walking patterns and movements. This makes it possible to stop potential dangers before they become problems. The efficiency of monitoring and surveillance is increased when gait analysis is integrated with surveillance

systems, such as CCTV cameras in public places. Security professionals can better respond quickly to any security problems or criminal activity by identifying suspicious or wanted individuals and evaluating the stride patterns of people seen on camera.

Gait analysis can be used to monitor crowd behavior and spot any indications of impending unrest or disturbance in crowded places like stadiums, airports, or public events. Security professionals can identify odd or violent conduct by observing the collective stride patterns and movements of people within a crowd. This enables them to take the necessary precautions to protect public safety and avert any possible threats.

Access control systems can improve identity verification by using gait analysis. Gait analysis adds another biometric factor for authentication and verification by identifying a person's specific gait pattern. This can be especially helpful in secure or limited settings where traditional identifying methods might not be appropriate or possible.

The examination of gait patterns in surveillance footage from crime sites can help forensic investigations. Investigators can determine the presence or absence of people at particular locations and perhaps connect them to illegal actions by comparing the stride patterns of suspects with those seen in video evidence. This helps to gather information and strengthen prosecution cases.

Security systems can use gait analysis to identify people with suspicious or hostile walking patterns, which enables security systems to proactively identify possible threats. Security staff are able to intervene before any detrimental activities take place thanks to this proactive strategy, which helps to either prevent or lessen the effects of security events.

## Crowd control

In a variety of situations, such as crowd-heavy stadiums, public gatherings, events, and public gatherings, crowd control is essential to protecting public safety. Gait analysis can help improve crowd control strategies by giving insightful information about crowd behavior and spotting potential hazards or disturbances.

Security officers can better comprehend crowd behavior by observing the collective stride patterns and movements of people within a crowd. Gait analysis can be used to identify atypical or combative gait patterns that may point to possible conflicts, combative conduct, or the presence of people who could endanger public safety.

A machine learning model can be trained to recognize particular gait patterns linked to unethical or aberrant behavior, and a trained model can be implemented in security systems to instantly identify and notify authorities about any crowds showing such behaviors by continuously observing the movement direction and crowd-flowing patterns of people. This makes it possible to intervene early and stops potential incidents before they get out of hand. This makes the security personnel respond quickly and specifically to any potential security risks or crowd disturbances.

During crowded events, gait analysis can assist security staff in more efficiently allocating their resources. Security officers can direct their attention to specific regions by detecting areas or groups of people where aberrant gait patterns are seen, resulting in a more focused and effective deployment of security personnel.

Gait analysis can be used to assess and analyze crowd behavior following an event in order to spot any security lapses or potential dangers that may have occurred. Authorities can learn more about crowd dynamics, identify people responsible for any disturbances, and enhance future crowd control tactics by evaluating CCTV footage and examining stride patterns.

It is significant to stress that privacy and legal factors should be taken into account when using gait analysis for crowd control. Adequate safeguards should be in place to guarantee the responsible application of surveillance technology while upholding the privacy rights of individuals and placing a priority on public safety.

### Emotion detection

Emotion detection is an important area of research in the field of artificial intelligence and computer vision. Traditionally, emotion detection has been performed using facial expressions or speech analysis. However, gait analysis has recently emerged as a promising method for emotion detection. One of the key advantages of using gait analysis for emotion detection is that it can be performed at a distance without the need for close physical proximity or the use of invasive sensors. This makes it a useful tool for applications such as surveillance, public safety, and crowd control.

Historically, speech analysis or facial expressions have been used to identify emotions. Gait analysis measures a range of walking-related characteristics that are connected to particular emotions. For instance, studies have shown that people typically take shorter steps and move more slowly when they are depressed, while they typically take longer steps and move more quickly when they are joyful. For instance, people who suffer from illnesses like melancholy, anxiety, or schizophrenia may walk differently, which might be a sign of their emotional state (*Feldman et al., 2020*). Gait researchers and medical experts may be able to recognize these diseases early and offer more precise therapies by examining gait patterns.

One of the main benefits of employing gait analysis for emotion detection is that it may be done remotely without invasive devices or near physical proximity. This makes it a practical tool for uses like crowd control, monitoring, and public safety. Emotion detection with gait analysis for surveillance and security is currently a relatively new field despite its potential benefits. There are a number of difficulties that must be overcome, such as the requirement for uniform measurement methods and the creation of precise algorithms for gait pattern analysis. In addition, ethical issues surrounding consent and privacy will need to be carefully taken into account, especially in the context of monitoring and public safety applications.

## DEEP LEARNING ARCHITECTURES IN GAIT ANALYSIS

While conventional machine learning and gait analysis techniques yielded reasonably acceptable outcomes in previous years, these methods often operated within the confines of manually designed features and exhibited limitations in capturing the underlying inherent patterns within data. Within this framework, the emergence of deep learning methodologies has offered a sophisticated avenue for addressing challenges in tasks

**Table 1  Comparison of neural network types.**

| Neural Network Type | Advantages | Disadvantages |
|---|---|---|
| CNNs | • Robust to spatial transformations<br>• Effective feature extraction<br>• High accuracy | • Require large datasets<br>• Computationally expensive<br>• Prone to overfitting |
| RNNs | • Model temporal dependencies<br>• Handle variable-length inputs<br>• Capture long-term dependencies | • Vanishing or exploding gradient<br>• Sensitive to hyperparameters<br>• Computationally expensive |
| Autoencoders | • Unsupervised learning<br>• Dimensionality reduction<br>• Anomaly detection | • Limited accuracy<br>• Sensitive to hyperparameters<br>• Lack of interpretability |
| GANs | • Generate realistic data<br>• Improve robustness<br>• Useful for data augmentation | • Training challenges<br>• Mode collapse<br>• Difficulty in handling temporal data |

**Table 2  Performance and efficiency of each type of neural network.**

| Neural network type | Accuracy | Training time | Inference time | Memory requirements | Scalability |
|---|---|---|---|---|---|
| CNNs | High | High | Medium | High | Good |
| RNNs | Medium | High | High | Medium | Poor |
| Autoencoders | Medium | Medium | Low | Low | Good |
| GANs | High | Very high | High | Very high | Poor |

involving images, videos, sequences, and more. Particularly in the domain of gait analysis, deep learning has proven to be a potent tool. Therefore, this section introduces the preeminent deep learning architectures that have been harnessed for gait recognition. Notably, these architectures encompass convolutional neural networks (CNNs), recurrent neural networks (RNNs), Autoencoders, and generative adversarial networks (GANs).

Each of these neural networks is discussed in terms of their strengths and weaknesses, with a summary provided in Table 1. Additionally, their performance and efficiency are compared in Table 2, focusing on metrics such as model accuracy, training time, inference time, memory requirements, and scalability. Accuracy is assessed with higher values indicating better performance, while training and inference times are critical for understanding the computational efficiency of each model. Memory requirements highlight the resources needed to process the network, and scalability reflects the ability of each network to handle large datasets.

## Convolutional neural networks

CNNs have revolutionized various fields, particularly image analysis and computer vision. Their effectiveness lies in their ability to automatically learn intricate features and hierarchies from raw data, making them especially well-suited for tasks such as image classification, object detection, and, notably, gait recognition.

At the core of CNNs is the convolutional layer, designed to mimic how the human visual system processes information. It applies convolutional operations to small regions of the input image, known as filters or kernels. These filters slide over the image, detecting different

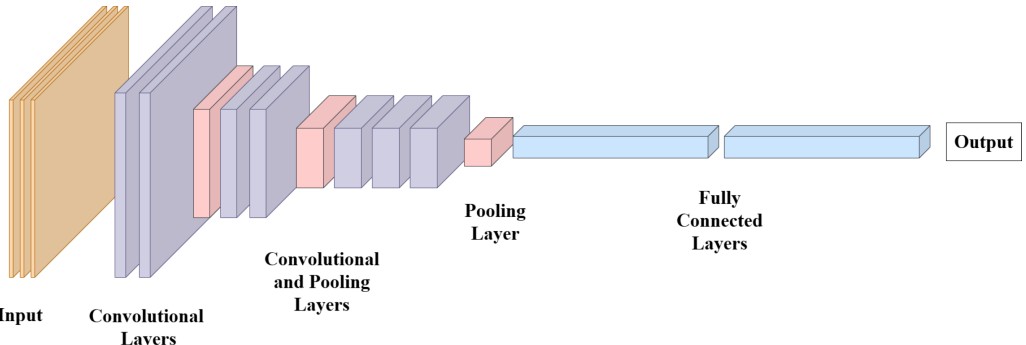

**Figure 4  Convolutional neural network architecture.**

features like edges, textures, and patterns. Through multiple layers of convolutions, the network can progressively learn complex features from simple to intricate, creating an abstract representation of the input data.

CNN architectures often consist of multiple layers, mainly convolutional layers, pooling layers, and fully connected layers, as depicted in Fig. 4. The input to a CNN is typically an image or a set of images represented as a multi-dimensional array. This array is received by the input layer which is set by parameters including the number of samples, height, width, and channels. The convolutional layer then extracts features from the array by applying filters to detect patterns and edges. The number of filters and filter size are key parameters of the convolutional layer that determine output feature maps. Other parameters, stride controls the step of the filter to move over the input array and padding determines whether to add values around the input to maintain the spatial dimensions.

Pooling layers downsample the data, reducing its spatial dimensions while preserving the learned features. This hierarchical feature extraction and downscaling allow CNNs to capture both local and global information from images effectively (*O'Shea & Nash, 2015*). Fully connected layers, also known as dense layers, are used at the end of the network to make predictions. The neurons in a fully connected layer receive input from all neurons in the previous layer and send output to all neurons in the next layer. The number of neurons in each layer can be defined as a parameter to increase the model to learn complex patterns. Moreover, dropout layers can be used to regularize the network by randomly dropping out neurons during training with setting dropout rate as parameter. Figure 4 depicts the general structure of CNN architecture.

While utilizing CNNs in gait analysis, a sequence of images and video frames of an individual's gait cycle is typically used as an input. The capability of learning and extracting relevant features of CNNs allows to capture gait features such as posture, and limb movements. By processing these features, CNNs can identify individuals and detect abnormality in gait.

Although CNNs have robustness to spatial transformations for images, they need large amounts of input data to train and are computationally expensive with significant resources

and time. Another drawback of CNNs can suffer is overfitting when there is small input data for training and poor performance in generalization.

## Recurrent neural networks

Following the discussion of CNNs, RNNs represent another specialized class of neural network architectures, particularly well-suited for handling sequences and time-dependent data. Their core ability is to maintain memory across time steps makes them particularly adept at tasks like natural language processing, speech recognition, and, notably, gait analysis.

At the core of an RNN is the concept of recurrent connections, where information from previous time steps is fed back into the network along with the current input. This mechanism enables RNNs to capture temporal dependencies and patterns within sequences, effectively learning from the order and context of data points (*Medsker & Jain, 2001*). The RNN layer maintains a hidden state that keeps information from previous time steps. At each time step, RNN takes the current input and the previous hidden states to calculate a new hidden state and output. The hidden state captures temporal dependencies to learn patterns from sequential data.

However, basic RNNs can struggle with capturing long-range dependencies due to a phenomenon known as the vanishing gradient problem. This limitation led to the development of more advanced RNN variants, such as long short-term memory (LSTM) networks and gated recurrent units (GRUs). These variants employ gating mechanisms to selectively retain and update information, enabling them to overcome the vanishing gradient issue and capture long-term dependencies more effectively.

One of the key benefits of RNNs is the ability to handle temporal dependencies which have variable-length inputs. Despite these advantages, RNNs have vanishing gradient problem due to long sequences of inputs. Another drawback of RNNs is that they are sensitive to hyperparameters. The parameters used in training such as selecting the number of hidden units and learning rate are needed to be chosen correctly to achieve the best performance. Similar to CNNs, RNNs can be computationally expensive according to the amount and complexity of input data.

In the context of gait analysis, RNNs can ingest sequences of gait data, such as time-series data of joint angles during walking including nuances of an individual's gait dynamics, accounting for variations in speed, rhythm, and phase. With alone RNNs, temporal information like time steps of an individual's gait can be used as input and by combining with CNNs, both spatial and temporal features can determine specific gait or identify individuals.

## Autoencoders

Apart from CNNs and RNNs, autoencoders offer a unique approach to neural network architectures, particularly in unsupervised learning, feature extraction, and data compression. Their fundamental purpose is to learn efficient representations of data by reducing its dimensionality while retaining its essential features. The core structure of an autoencoder consists of two main components: an encoder and a decoder. The

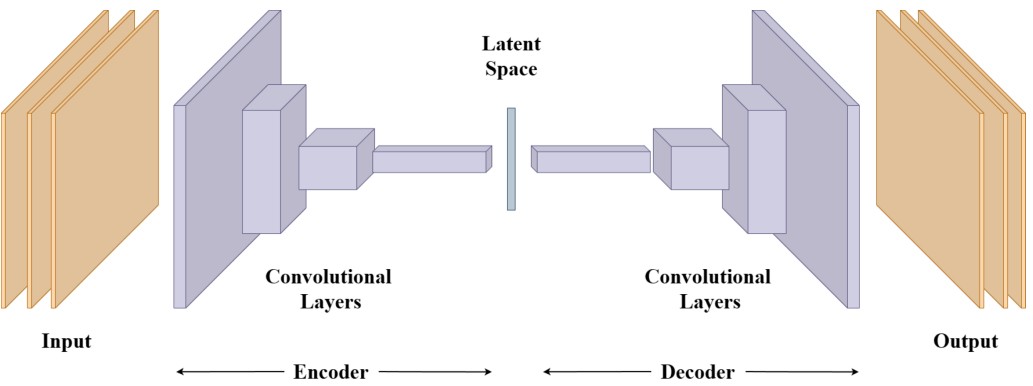

**Figure 5  Autoencoder architecture.**

encoder transforms the input data into a lower-dimensional representation, often referred to as a latent space or bottleneck. This process involves compressing the input data into a compact format that captures its most important characteristics. The decoder then aims to reconstruct the original data from this lower-dimensional representation. The learning process involves minimizing the difference between the input data and its reconstructed output. Figure 5 shows the general architecture of autoencoders.

In Fig. 5, the encoder and decoder are depicted as a series of convolutional layers. These layers consist of convolutional layers that extract features from the input data and can be followed by activation functions and pooling operations to reduce or expand the spatial dimensions and increase the feature depth. Specific parameters used in the convolutional layer like the number of filters, filter size, stride, and padding can be used to determine the model's performance.

The learning process involves minimizing the difference between the input data and its reconstructed output. By doing so, the autoencoder forces itself to learn a compact and informative representation of the input data. This latent space can serve as a feature extraction mechanism, enabling the autoencoder to capture intricate patterns that are not immediately apparent in the raw data (*Kingma & Welling, 2019*).

In gait analysis, autoencoders can be particularly valuable in scenarios where data might be noisy, incomplete, or variable due to factors like different clothing or camera angles. Autoencoders' compressed latent space contains only essential gait features and, excludes noisy and invaluable information. This learning process makes autoencoders easier to work and reduces computational requirements. Moreover, from lower dimensional representation or latent space, autoencoders can generate synthesized gait patterns like human poses.

Autoencoders can capture essential features from input which makes them useful for unsupervised learning and removing unnecessary noise from data. On the other hand, the architecture and hyperparameters of the model have to be chosen wisely to capture important features. The latent space in the architecture is difficult to understand and hard to interpret.

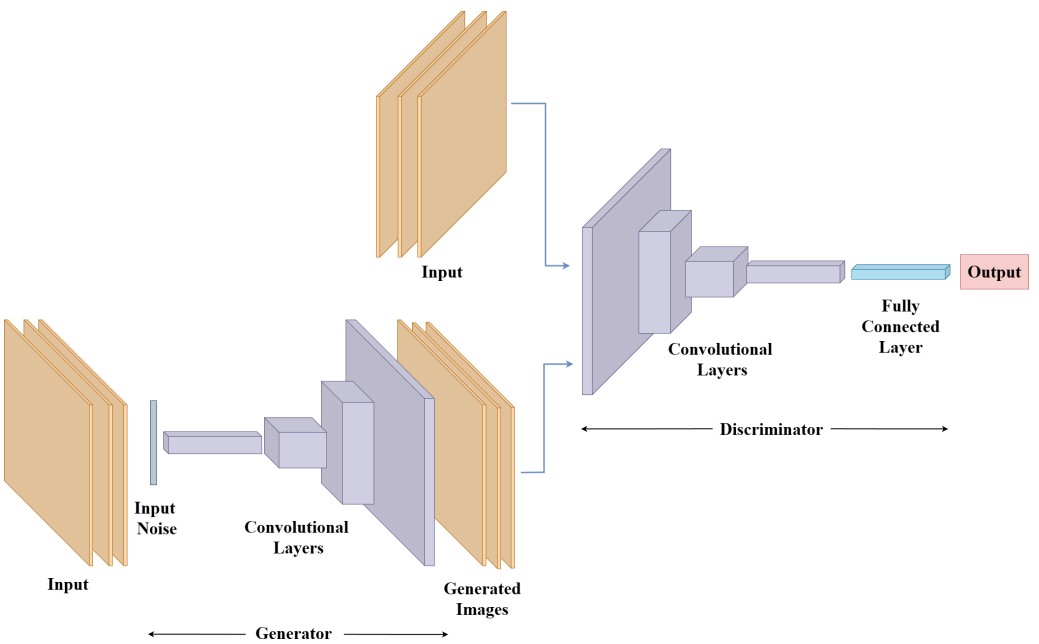

**Figure 6  Generative adversarial network architecture.**

## Generative adversarial networks

Finally, GANs represent a groundbreaking approach within deep learning, particularly well known for their ability to produce synthetic data that closely resembles real-world samples. The GAN framework consists of two integral components: the generator and the discriminator. This unique learning process generates synthetic output from input data which the generator analyzes to produce new data, while the discriminator is used to distinguish between real and generated samples.

The generator's task is to synthesize data that closely resemble genuine examples. It starts with random noise and gradually refines its output through successive layers, mimicking the patterns and characteristics present in the training data. In Fig. 6, the generator receives noised images and generates synthetic images that resemble the real input images using convolutional layers. The discriminator, on the other hand, receives both real and synthetic images. The convolutional layers in the discriminator learn features from both inputs and the fully connected layer acts as a classifier to assess the similarity between the synthetic image and the real image. As training progresses, the generator becomes increasingly adept at producing realistic samples, while the discriminator fine-tunes its ability to detect accurate distinctions (*Creswell et al., 2018*). As for structural parameters for GAN, input size, and shape can be defined for the images and the number of filters, filter size, stride, and padding can be specified for convolutional layers used in the generator and discriminator. The general architecture of GAN is depicted in Fig. 6.

GANs can provide data augmentation and generate more data for training which makes them able to handle various conditions like different lighting or occlusions for gait recognition, in which synthetic data preserves privacy by not revealing the identities of the

individuals. However, the generator may only generate a limited variation of data due to mode collapse and the generated synthetic data can be inaccurate.

# DEEP LEARNING BASED GAIT ANALYSIS APPROACHES

Gait represents an individual's unique walking pattern. Therefore, it can be applied as a biometric identifier for identification purposes. In comparison to other identification methods using face, iris, and fingerprints, gait holds significant advantages. For instance, facial recognition may fail to identify an individual from a distance or in low-resolution scenarios. Moreover, certain studies have shown that other identification methods can be falsified or substituted, whereas it is challenging for an individual to imitate their gait due to the restrictions it places on their movement. For these reasons, person identification or re-identification through gait analysis offers several advantages. Furthermore, it does not require the subject's cooperation, can be performed from a distance, is not easily concealed, and is unique to each person. However, gait-based identification also has limitations, such as potential changes in walking patterns influenced by factors like age, illness, and emotional states.

Gait data can be obtained from cameras, inertial sensors, and pressure sensors. However, pressure and inertial sensors are less likely to be employed for the purpose of identification compared to video footage from cameras. This is because using sensors sometimes requires consent from the subject. Therefore, gait analysis using video from cameras gains significant advantages since it can be performed from a distance without the knowledge of the subject. However, consent would be required if the analysis is conducted by authorized agents such as police and security agencies.

## Gait recognition

Gait recognition is becoming a popular human identification technique in several areas, such as security, medical examination, biometrics, access control, identity management, and so on. Other biometrics, such as fingerprints, can be faked, and facial recognition has some inaccuracy issues (*Avarur et al., 2023*). Compared to other biometrics, gait recognition has better advantages in its own way. Because it can be used in dim light and long distances with less clothing hindrance, suspects who are trying to prevent and hide other biometrics easily but gait patterns would not prevent them from revealing their true identities (*Velapure & Talware, 2020*).

Due to the consecutive wave of technologies, several techniques have been applied for gait recognition. For example, model-based, template-based, and machine learning-based approaches. The model-based method is the assumption and description of a full gait cycle from a limited number of gaits. The template-based method includes matching the new strides to the pre-existing ones. The machine learning approach comprises feeding a large number of gait sets and patterns to a computer machine, where it learns the features and generates a model. The generated model can later be used to recognize the structure and pattern of human beings. To process a huge amount of gait datasets, deep learning approaches such as CNN, RNN, GAN, and Encoder/Decoder-based machine learning algorithms are becoming popular for gait recognition (*Pundir & Sharma, 2023*).

### Model free approaches for gait recognition

Deep learning-based gait recognition approaches are now becoming one of the most interesting research areas for person identification. According to the ability to learn different features from huge amounts of raw data with deep learning techniques, model-free-based approaches have been proven to make significant promises for gait recognition, which learns patterns from color formation and motion in images and video frames. However, model-free approaches draw biases on appearance, such as texture and colors. The following are some researches that have been carried out: gait recognition using appearance-based or model-free approaches during recent years.

The study by *Hasan & Mustafa (2020)* focuses on the difficult task of gait recognition, specifically tackling the issues posed by changes in illumination and clothing. This research introduces a simple yet effective approach using an RNN with a GRU architecture. This RNN captures the temporal dynamics of body pose sequences for recognition. A low-dimensional gait feature descriptor is designed based on 2D coordinates of human pose data. This descriptor is not only invariant to various factors but also captures gait pattern dynamics. Experimental results on challenging gait datasets demonstrate that this method achieves state-of-the-art performance in both single-view and cross-view gait recognition, showcasing its effectiveness.

In addition, *Kusakunniran et al. (2010)* addressed the challenge of obtaining reliable gait features for identification when there are changes in viewing angles. Gait can vary in appearance at different angles. To solve this, the article proposed the problem as a regression task and introduced a novel View Transformation Model(VTM). The proposed VTM was constructed using a multi-layer Perceptron (MLP) as a regression tool and estimated gait features for an unknown viewing angle using motion information from a well-defined Region of Interest (ROI) under other known viewing angles. It also normalized gait features across diverse viewing angles, enabling consistent gait similarity measurements.

As mentioned in the previous paragraphs, numerous studies have been conducted in the field of gait recognition using model-free techniques. The integration of gait recognition and model-free approaches offers numerous advantages. Using deep learning techniques, their ability to recognize complex and detailed patterns enhances gait recognition's accuracy by capturing even small differences in an individual's walking style. This adaptability extends to varying real-world conditions, where factors such as clothing changes and different lighting may arise as deep learning models generalize effectively. Additionally, they can autonomously extract relevant features from raw gait data, reducing the need for manual feature engineering. The model's capacity to continuously refine its understanding over time ensures a progressive improvement in recognition performance. Furthermore, deep learning based gait recognition researches are summarized with applied deep learning models and their performances in Table 3 and the datasets in Table 4.

### Silhouette based approaches in gait recognition

The process of using silhouettes holds importance in numerous computer vision tasks, especially in person identification. In the extraction of silhouettes, the detected body is separated from the background imagery, resulting in a binary silhouette representation of

**Table 3   Deep learning based gait recognition articles with their performances.**

| Ref. | Dataset | Model | Metric | Performance score |
|---|---|---|---|---|
| Fan et al. (2023) | CASIA B, OU-MVLP, GREW | CNN | Accuracy | 92% |
| Chen et al. (2023) | CASIA-B, OU-MVLP | CNN | Accuracy | 98.6% |
| Huang et al. (2022) | CASIA B, OU-MVLP | CNN | Accuracy | 97.3% |
| Hou et al. (2021) | CASIA B, OU-MVLP | RNN, CNN | Accuracy | 97.1% |
| Zhao et al. (2021) | whuGAIT, OU-ISIR | LSTM | Accuracy | 94.15% |
| Xu et al. (2020) | OU-MVLP, CASIA-B | GAN | Identification rate | 93.2% |
| Wang & Yan (2020) | OU-MVLP, OU-LP, CASIA B | CNN | Identification rate | 98.93% |
| Zou et al. (2020) | whuGAIT | CNN + LSTM | Accuracy | 99% |
| Babaee, Li & Rigoll (2019) | CASIA A, CASIA B | GAN | Recognition rate | 82% |
| Zhang et al. (2019) | OU-ISIR, CASIA B | Autoencoders | Accuracy | 96.15% |
| Wang et al. (2019) | CASIA B, USF, FVG | Autoencoders | Accuracy | 99.1% |
| He et al. (2018) | OU-ISIR, CASIA B, USF | GAN | Accuracy | 94.7% |
| Yu et al. (2017) | CASIA B, SZU RGB | Autoencoders | Identification rate | 97.58% |

**Table 4   Existing datasets for gait recognition.**

| Ref. | Dataset name | No. of subjects | No. of sequences | Environment |
|---|---|---|---|---|
| Topham et al. (2023) | – | 64 | 3,120 | Indoor, Outdoor |
| Song et al. (2022) | CASIA-E | 1,014 | 778,752 | Outdoor |
| Zheng et al. (2022) | GAIT3D | 4,000 | 25,309 | Outdoor |
| Zhu et al. (2021) | GREW | 26,000 | 128,000 | Outdoor |
| Mu et al. (2021) | ReSGait | 172 | 870 | Indoor |
| Nunes, Moreira & Tavares (2019) | GRIDDS | 35 | 350 | Indoor |
| Takemura et al. (2018) | OU-MVLP | 10,307 | 267,386 | Indoor |
| Mansur et al. (2014) | OUISIR Speed Transition | 179 | – | Indoor |
| Hofmann et al. (2014) | TUM-GAID | 35 | 1,645 | Indoor |
| Zheng et al. (2012) | CASIA-dataset D | 88 | 880 | Outdoor |
| Makihara et al. (2012) | OU-ISIR, Treadmill | 32 | 2,746 | Indoor |
| Iwama et al. (2012b) | OUISIR Large Dataset (OULP) | 4,007 | – | Indoor |
| Nixon, Tan & Chellappa (2010) | SOTON, Small Dataset | 12 | – | Indoor |
| Tan et al. (2006) | CASIA-dataset C | 153 | 1,530 | Outdoor |
| Yu, Tan & Tan (2006) | CASIA-dataset B | 124 | 13,640 | Indoor |
| Shutler et al. (2004) | SOTON, Large Dataset | 115 | 2,128 | Indoor, Outdoor |
| Liu et al. (2004) | USF | 122 | 1.970 | Outdoor |
| Wang et al. (2003) | CASIA-dataset A | 20 | 240 | Outdoor |
| Little & Boyd (1998) | UCSD | 6 | 42 | Outdoor |

the body. With the extraction of a silhouette image, diverse techniques for feature extraction can be employed for gait analysis. These include methods like Gait Energy Image (GEI), Depth-GEI (which involves depth data for improved quality), Gait Energy Volume (GEV), and Depth Gradient Histogram Energy Image (DGHEI). The GEI technique assumes that

a comprehensive representation of gait information can be captured within a single gait cycle. This approach, while potentially eliminating noise, might also eliminate vital details. Notably, generating silhouettes solely from RGB data can result in suboptimal quality, potentially failing to accurately outline the body (_Hasan & Alani, 2022_). In contrast, Depth-GEI emerges as an alternative, mitigating these shortcomings by utilizing depth data to yield higher-quality GEIs. Another avenue, GEV, expands upon GEI, aggregating three-dimensional binary voxel volumes (_Ramakić et al., 2020_). Another notable method, DGHEI, adopts the essence of GEI, averaging feature vectors of each frame within a gait cycle. Additionally, DGHEI incorporates the edges and depth gradients accessible, in-depth data (_Sakata et al., 2019_).

Another hindrance in gait recognition is variance in carrying an object, which impacts an individual's gait and affects identification with gait. In this article by _Ghosh (2022)_, the authors addressed the issue of identifying individuals with or without carrying objects. The proposed method utilized a Faster Region Convolutional Neural Network (FR-CNN) by modifying it to detect and extract pedestrians from video frames, regardless of whether they are carrying objects. Then, Deep convolutional layers generated feature vectors from pedestrian movements across frames. These feature vectors undergo analysis using two versions of RNN: LSTM and bidirectional long-short term memory (BLSTM). The performance of the proposed system is evaluated on four public datasets: OU-ISIR Large Population Gait database with real-life carried objects (OU-LP-Bag), OU-ISIR Gait database Treadmill dataset B (OUTD-B), OU-ISIR Large Population Gait database with Age (OULP-Age), and CASIA Gait database B(CASIA-B). Experimental results validated that this proposed gait recognition system outperforms other existing state-of-the-art methods.

In the work by _Babaee, Li & Rigoll (2019)_, the authors suggested a method to identify individuals even if there are limited or few gait patterns available. They proposed a full CNN model that can generate a complete GEI from an incomplete one. They trained auto encoders independently and then re-united these into the united model. The proposed model was experimented on two datasets, OULP and CASIA-B. The result showed that their proposed model can successfully reconstruct GEI and improve the rate of gait recognition speed. However, the suggested model can only regenerate GEI from side-view, and their future work will extend to process gait recognition of cross-view. In the meanwhile, _Wang, Zhang & Yan (2020)_ presented a novel gait recognition scheme to tackle the issue of cross-view gait detection. First, they designed a new gait feature representation called triple gait silhouettes, generated from successive gait silhouette images. Then, a multichannel CNN network was constructed to process the consecutive silhouette images in parallel. Their proposed model was evaluated on the CASIA A/B dataset for cross-view gait recognition and the OU_ISIR dataset to testify to their model capability with the huge dataset.

### Model based approaches in gait recognition

It can be noticed that the outline of the binary silhouette images can be easily affected by clothing and accessories. Addressing this challenge, an alternative avenue involves adopting the model based approach, which strives to establish a consistent skeleton model

irrespective of the attire or items the individual is adorned with. The movement patterns of human skeletons inherently hold substantial insights into the subject's activities, gait, and identity. Executing the skeleton-centric approach necessitates the extraction of an individualized skeleton for each person through precise pose estimation techniques.

The research work done by *Zheng et al. (2022)* addresses the limitations of current gait recognition studies that predominantly use 2D representations in controlled settings. Recognizing that humans move in unconstrained 3D environments, the article proposes a novel approach named "SMPLGait" for gait recognition using dense 3D representations. The framework leverages the 3D Skinned Multi-Person Linear (SMPL) model to capture essential information like viewpoint, shape, and dynamics. The approach consists of two branches—one extracting appearance features from silhouettes and the other learning 3D information from the SMPL model. To support this, the article introduces the Gait3D dataset, a significant collection of 3D representation-based gait data extracted from multiple cameras in uncontrolled indoor environments. The dataset provides detailed 3D SMPL models that encapsulate body shape, viewpoint, and dynamics. Evaluating the proposed method against existing approaches using Gait3D data demonstrates the superiority of the framework and highlights the potential of 3D representations for gait recognition in real-world scenarios.

Moreover, the study by *Gao et al. (2019)* introduces a novel approach for human gait analysis using a wearable inertial measurement unit (IMU). The system focuses on identifying abnormal gait patterns, such as hemiplegic, tiptoe, and cross-threshold gaits. It employs the dynamic step conjugate gradient algorithm to compute gait data attitude and a gait feature information location algorithm for attitude data segmentation. These segmented data are then fed into a classification model based on a long short-term memory network and convolutional neural network (LCWSnet). The algorithm adapts feature parameters based on objectives and optimization functions, enhancing the LSTM-CNN model's convergence layer to improve abnormal gait classification accuracy. Experimental results affirm the efficacy of the LCWSnet-based method in detecting abnormal gait patterns, showcasing its accuracy and potential for practical implementation.

## Reidentification

Reidentification (ReID) is a computer vision task that involves matching individuals across different images or video frames. It aims to identify and track the same person across different cameras or instances. Gait-based ReID, specifically, focuses on utilizing the unique characteristics of a person's walking style or gait for identification purposes. The operation of recognition of an individual from a set of images or video footage captured by several cameras is called person reidentification. In this process, similarity is the key to computing the matching between two or a set of images (*Gao et al., 2016*).

The sequential methods that are used to list the features of a person are not efficient for the reidentification of a person due to several limitations, such as differences between the analyzed objects in terms of colors, scales, shapes, and others. The use of limited features cannot be enough for proper identification. Whereas, by using deep machine learning techniques, the use of different and non-limited features for learning becomes

another alternative for solving the problem of person reidentification. However, to train the machine learning model, a huge amount of data is required from multiple video footage. In addition, machine learning performance can be enhanced by using preprocessing techniques. Collecting multiple images of the same person captured by different cameras can be difficult. The difficulties can be in the shape, the color of the clothes as well as the scale of the images. In such a case, gait is one of the features that cannot be changed and is repetitively performed by the person; this could be considered as an identification feature (*Wu et al., 2018*).

For instance, *Sattrupai & Kusakunniran (2018* introduced a novel solution utilizing dense trajectory analysis, commonly used in action recognition, to extract gait information. In this approach, dense trajectories were employed to detect key points and their corresponding trajectories. From each key point, descriptors like HOG, HOF, MBHx, MBHy, and dense trajectory are extracted, forming the point descriptor. During the training phase, a bag of words (BoW) model was trained using point descriptors from training gait videos. Subsequently, in the testing phase, the BoW model was used to extract gait features from each video, forming the gait feature. The experimental results conducted on the CASIA gait database B demonstrated the promising performance of the proposed method across various viewing angles. This technique falls within the domain of gait recognition, focusing on feature extraction and representation, rather than traditional ReID (person reidentification) which, often involves matching individuals across different images or videos.

Moreover, *Jin et al. (2022)* addressed the challenge of Cloth-Changing Person Reidentification (CC-ReID), aiming to match individuals across different locations and times while accounting for clothing changes. The focus is on efficiently performing person identity matching using a single image, which is especially suited for surveillance scenarios. The proposed approach, named GI-ReID, introduced gait recognition as an auxiliary task to enhance cloth-agnostic representations in an Image ReID model. GI-ReID employed a two-stream architecture with an Image ReID-Stream and an auxiliary Gait-Stream. The Gait-Stream served as a regulator during training, promoting the capture of cloth-invariant biometric motion features in the ReID-Stream. To incorporate temporal motion cues from a single image, a Gait Sequence Prediction (GSP) module was designed for the Gait-Stream. Additionally, a semantics consistency constraint was applied to both streams for effective knowledge regularization. The experiments conducted on multiple benchmarks related to image-based Cloth-Changing ReID, including LTCC, PRCC, Real28, and VC-Clothes, show that GI-ReID outperforms state-of-the-art methods, demonstrating its effectiveness in addressing the challenges of CC-ReID.

Furthermore, *Rao et al. (2021)* presented person reidentification (Re-ID) through gait features extracted from 3D skeleton sequences, presenting a novel self-supervised approach for learning gait representations in the absence of labeled data. Unlike existing methods that rely on hand-crafted descriptors or supervised learning, this approach leverages unlabeled skeleton data for gait representation learning. The proposed self-supervised framework introduced reverse sequence reconstruction as a form of self-supervision, utilizing richer high-level semantics to enhance gait representation learning. Additionally, the study

**Table 5  Deep learning based pose estimation articles with performance scores.**

| Ref. | Dataset | Model | Metric | Performance score |
|------|---------|-------|--------|-------------------|
| *Huang et al. (2023)* | MPII, COCO, CEPDOF WEPDTOF Pose, BKFisheye | CNN | AP | 90.7% |
| *Jiang et al. (2023)* | COCO, AP-10K CrowdPose, MPII Human Pose | CNN | AP | 68.7% |
| *Zheng et al. (2023)* | COCO | CNN | AP | 74.9% |
| *Xu et al. (2022)* | COCO | CNN | AP | 65.4% |
| *Wang et al. (2022)* | COCO, CrowdPose | CNN | AP | 58.3% |
| *Cai et al. (2020)* | COCO, MPII Human Pose | CNN | mAP | 79.2% |
| *Zhang et al. (2020)* | COCO, MPII Human Pose | Encoder-Decoder | mAP | 76.2% |
| *He et al. (2017)* | COCO | CNN | mAP | 71.4% |
| *Huang et al. (2020)* | COCO | Encoder-Decoder | mAP | 76.5% |
| *Sun et al. (2019)* | COCO, MPII Human Pose | CNN | mAP | 77% |
| *Osokin (2018)* | COCO | CNN | mAP | 42.8% |
| *Cao et al. (2017)* | COCO, MPII Human Pose | CNN | mAP | 75.6% |

explored other pretext tasks to further enhance self-supervised learning. To capitalize on the temporal continuity inherent in adjacent skeletons and consecutive skeleton sequences, a locality-aware attention mechanism and a locality-aware contrastive learning scheme were introduced. These mechanisms aim to preserve locality-awareness within and between sequences during self-supervised learning. Finally, leveraging context vectors learned from the attention mechanism and contrastive learning scheme, a new feature called Contrastive Attention-based Gait Encodings (CAGEs) was devised for effective gait representation. Empirical evaluations demonstrated that this approach outperforms existing skeleton-based methods by a significant margin in terms of Rank-1 accuracy (15-40 percent improvement), even outperforming various multi-modal methods that incorporate additional RGB or depth information.

## Pose estimation

Pose estimation is the process of estimating the human body's position and orientation. It has significant applications in gait analysis and could be used in criminal investigations. In gait analysis, pose estimation is used to extract biomechanical features, including joint angles and trajectories, which can be used to identify individuals based on their unique walking patterns or gait. By analyzing the movement of various body parts, such as the legs, arms, and torso, pose estimation algorithms can create a model of an individual's gait, which can be used to identify them in surveillance footage or crime scene videos. The summary of pose estimation articles with performance scores is provided in Table 5, and the datasets for pose estimation in Table 6.

In gait analysis, pose estimation can provide valuable information about a person's walking patterns, such as stride length, step width, and cadence. By analyzing these parameters, researchers can identify abnormalities in a person's gait, which can be indicative of various neurological or musculoskeletal conditions. In forensic investigations,

**Table 6** Existing datasets for pose estimation.

| Ref. | Dataset name | Number of records | Labelling |
|---|---|---|---|
| *Huang et al. (2023)* | BKFisheye | 49,578 | Keypoints |
| *Tezcan et al. (2022)* | WEODTOF-Pose | 10,544 | Keypoints |
| *Duan et al. (2020)* | CEPDOF | 25,504 | Keypoints |
| *Von Marcard et al. (2018)* | 3DPW | – | 3D poses |
| *Xia et al. (2017)* | PASCAL-Person-Part | 3,533 | Keypoints and body segments |
| *Andriluka et al. (2014)* | MPII Human Pose | 40,522 | Keypoints and activities |
| *Lin et al. (2014)* | Common Objects in Context (COCO) | 200,000 | Keypoints |
| *Ionescu et al. (2013)* | Human3.6M | 3,600,000 | 3D poses |

pose estimation can be used to identify suspects or victims based on their unique walking patterns (*Kasprzyk, 2023*). By comparing the gait of an individual in surveillance footage or crime scene videos to a database of known gait patterns, law enforcement officials can potentially identify suspects or victims with a high degree of accuracy.

There are several approaches to pose estimation in gait analysis for crime investigation, including marker-based and marker-less methods. Marker-based methods involve placing markers on specific body parts and tracking their movement over time. While marker-based methods can provide accurate measurements of body movement, they require extensive calibration and can be cumbersome to use in real-world scenarios. Marker-less methods, on the other hand, use computer vision algorithms to automatically detect and track body parts without the need for markers. While marker-less methods are generally easier to use, although they may not provide the same level of accuracy as marker-based methods, they are suitable for analyzing surveillance videos that involve criminal activities and images from crime scenes.

### 2D human pose estimation

2D human pose estimation (HPE) is a technique used to estimate the pose of a person in an image or video. In criminal investigation, it can be useful for identifying and tracking suspects, analyzing their movements and behavior, and determining their involvement in a crime. With 2D human pose estimation, the positions of key points on the human body, such as the head, shoulders, elbows, hips, and knees, can be estimated from an image or video frame. These points are then used to create a skeleton-like representation of the person's pose. This can be helpful for detecting abnormal or unusual movements or postures that might be indicative of criminal activity. For instance, if a suspect is caught on camera walking in a specific way, the 2D human pose estimation algorithm can help law enforcement officials identify and analyze that gait. This could potentially help investigators identify and track a suspect more accurately. Another application of 2D human pose estimation in criminal investigation is in identifying potential witnesses or suspects. By analyzing video footage from a crime scene or public area, investigators can use the algorithm to detect and track individuals based on their pose and movements. With

the help of deep learning methods and applications in HPE, it provides the ability to learn feature representations automatically from video and image data.

Within the domain of 2D HPE, the pursuit of an algorithm that seamlessly balances precision and efficiency becomes paramount in crafting a comprehensive human representation and capturing the intricacies of the subject's movement (*Zheng et al., 2020*). The attainment of high-precision detection is pivotal as it underpins the precise extraction of human body information, consequently enriching subsequent tasks like 3D HPE and action recognition. Nevertheless, several challenges loom in the path towards achieving this level of accuracy, stemming from various sources. Firstly, real-world scenes frequently introduce nuisance factors like under/over-exposure and the intertwining of humans with objects, considerably amplifying the risk of detection failures in HPE. Secondly, the inherent flexibility of human kinematic chains leads to instances of pose occlusions, including self-occlusions in numerous scenarios, inevitably confounding keypoint detectors reliant on visual cues. Thirdly, the common occurrences of motion blur and video defocus in video sequences further exacerbate the challenge, ultimately eroding the precision of HPE outcomes (*Josyula & Ostadabbas, 2021*).

Pose estimation of the individuals from video footage captured by surveillance cameras has gained significant attention in recent research. Processing video footage always brings several challenges, including varying camera angles, focus, and positions, all of which can impact video quality. Despite these hindrances, surveillance camera videos offer valuable clues and an abundance of information for pose estimation. To tackle analyzing videos for pose estimation, temporal information must be considered, as applying the existing methods of training on static images might lead to unsatisfactory results in processing video frames.

The Optical Flow model captures the movement of individual pixels across frames, and through the identification of changing pixels between frames, it becomes possible to deduce pose estimations. A synthesis of convolutional networks and optical flow within a unified framework is accomplished by *Pfister, Charles & Zisserman (2015)*, utilizing the flow field to temporally align features across multiple frames. This alignment of features is then harnessed to enhance the precision of pose detection within individual frames. Another notable contribution by *Song et al. (2017)* introduces the Thin-Slicing Network, which computes comprehensive optical flow between sequential frames. This calculated flow aids in propagating the initial estimation of joint positions across time. This network further employs a flow-based warping mechanism to align joint heatmaps, a procedure that lays the foundation for subsequent spatiotemporal inference.

However, some challenges persist in existing methods that rely on optical flow. While optical flow contains beneficial features such as human motion details, it also involves unwanted background changes. The inclusion of noisy motion representation considerably hinders these proposed methods from achieving the desired accuracy. The optical flow-based representation effectively captures motion cues at a pixel level, which is advantageous for capturing essential temporal information. Nevertheless, optical flow has limitations, as it tends to extract impure features and is particularly susceptible to noise interference.

### Top down approach

In the top down approach of pose estimation, it begins with the detection of human subjects in the input image or video. Once the human subjects are detected and localized with bounding boxes, the next step is to predict the key points or joints of the detected human subjects within these bounding boxes. These key points represent important anatomical landmarks, such as shoulders, elbows, wrists, hips, knees, and ankles.

The top-down approach follows a two-step process, where the first step is the detection of human instances using object detection models like Faster R-CNN (*Ren et al., 2015*) or Mask R-CNN (*He et al., 2017*), which are designed to locate and classify individual objects within the image. The second step involves predicting the key points using specialized pose estimation models, typically based on deep learning architectures like CNN. These pose estimation models take the localized human instances as input and output the coordinates of the key points on each individual.

The body pose representation is constructed by connecting the identified key points. This linkage between key points creates a structural representation of the individual's posture and body position. This representation is valuable in analyzing human movements, gestures, and interactions. However, the computational time required for this process is influenced by the number of individuals present in the image or video frames. As the number of individuals increases, the time to perform person detection and key point identification rises accordingly.

In the context of criminal investigation, this approach can provide valuable insights and evidence. For example, in cases involving suspicious individuals captured on surveillance footage, pose estimation can help law enforcement agencies analyze body movements and gestures, potentially revealing hidden intentions or behavioral patterns. Moreover, it can aid in identifying specific suspects using gait recognition from extracted gait poses and movements, especially when their faces are obscured or disguised. To analyze gait with this approach, accurate pose estimation is challenging in crowded or cluttered environments where occlusions and overlapping subjects occur, and this relies on the quality and resolution of the input data, as well as the performance of the applied pose estimation models.

### Bottom-up approach

In the bottom-up approach of pose estimation, key points are initially detected and identified across the entire image or video frame without the need for prior detection of human subjects. These key points are then grouped together to form estimated poses for each person present in the scene. Firstly, the model detects all possible key points, regardless of whether they belong to the same person or not. After predicting key points, the next step involves grouping them to form complete body poses by establishing associations between key points and linking those that are likely to belong to the same person. For gait analysis, *Kusakunniran (2015)* detected body parts such as the head, waist, knees, and ankles to develop human posture from sequences of gait images.

First, the videos are processed to extract gait-related features (*Kusakunniran, 2015*), such as step length, stride length, step width, cadence, posture, and body movements. These

features are then analyzed and compared with established norms or reference datasets to determine if any abnormalities are present.

This approach is particularly beneficial when dealing with scenarios involving occlusions or interactions between individuals where they may partially block each other in the image and keypoints can still be detected and associated correctly, even when certain body parts are obscured or hidden from view. The other advantage of this approach lies in its reduced computing time and power consumption compared to the top-down approach, as it avoids the need for separate keypoint detection for each individual. This efficiency makes it particularly well-suited for scenarios with multiple subjects or crowded scenes.

For criminal investigation, the bottom-up approach detects key points in a more robust and efficient manner, even in crowded and complex scenarios, which is more suitable for analyzing low-quality or resolution video frames from surveillance cameras. For instance, in cases involving altercations or suspicious activities captured in surveillance footage, the bottom-up pose estimation can help identify crucial key points, such as hand gestures or body postures, that may provide valuable clues about the intentions or actions of the involved parties. Moreover, this approach can assist in the reconstruction of human subjects, although they are partially occluded or obscured, in which investigators may gain insights into the sequences of human movements and the dynamics that occurred during the incident.

## PRACTICAL APPLICATIONS AND RESEARCHES ON GAIT ANALYSIS FOR CRIMINAL INVESTIGATION

Gait analysis using deep learning has found several practical applications in real-world scenarios within criminal investigations and forensic science. One significant application is suspect identification from surveillance footage. In situations where a suspect's face is obscured or the video quality is too low for facial recognition, law enforcement agencies have turned to gait analysis to identify suspects based on their walking patterns. Deep learning models trained on gait data can match a suspect's gait captured in low-quality surveillance footage with a database of known individuals, providing crucial leads in investigations.

Another important application is tracking and monitoring individuals. In cases requiring continuous monitoring of a group of people, gait analysis can track individuals' movement across multiple camera views, even when their faces are not visible. This technique has proven particularly useful in monitoring large public spaces, such as airports or train stations, where maintaining continuous surveillance is challenging due to the scale and complexity of the environment. In mass surveillance tracking and monitoring scenarios, such as during large public gatherings, protests, or riots, gait analysis offers a non-intrusive method to identify and track individuals within a crowd (*Xijuan, Yusoff & Yusoff, 2022*).

Gait analysis also plays a role as forensic evidence in court. By analyzing the gait of individuals captured on video at a crime scene, experts can present gait patterns as a form of biometric evidence. This supplementary evidence can be crucial in supporting the identification of suspects, especially when other forms of evidence are unavailable (*Liu et*

**Table 7  Criminal activity datasets.**

| Ref. | Dataset name | No. of classes | Type of classes | No. of videos |
|---|---|---|---|---|
| *de Paula, Salvadeo & de Araujo (2022)* | CamNuvem | 1 | Robbery | 486 |
| *Kwan-Loo et al. (2022)* | Kranok-NV | 2 | Violent and non-violent activities | 3,683 |
| *Cheng, Cai & Li (2021)* | RWF-2000 | 2 | Violent and non-violent activities | 2,000 |
| *Öztürk & Can (2021)* | UCF Crime Extension | 3 | Normal activities and criminal activities | 240 |
| *Boekhoudt et al. (2021)* | HR-Crime | 14 | Normal activities and criminal activities | 950 |
| *Wu et al. (2020)* | XD-Violence | 6 | Normal activities and criminal activities | 4,754 |
| *Perez, Kot & Rocha (2019)* | Real-world Fight | 2 | Violent and non-violent activities | 1,000 |
| *Aktı, Tataroğlu & Ekenel (2019)* | Fight detection Surv Dataset | 2 | Violent and non-violent activities | 1,000 |
| *Garje, Nagmode & Davakhar (2018)* | SDHA 2000 | 2 | Violent and non-violent activities | – |
| *Sultani, Chen & Shah (2018)* | UCF Crime | 13 | Normal activities and criminal activities | 1,900 |

*al., 2021*; *Lin et al., 2021*). In addition, automated anomaly detection has become a valuable tool in surveillance and public safety. Deep learning-based gait analysis can automate the process of detecting unusual or suspicious gait patterns and actions.

Another use case is application in cold case investigations, which are long-unsolved cases, archived as surveillance videos and images, and contain overlooked evidence due to the technological limitations at the time (*Liu et al., 2021*; *Lin et al., 2021*). With advanced deep learning models, this evidence can be analyzed to identify perpetrators or suspects who were previously missed.

Integrating deep learning-based gait analysis into surveillance and criminal investigations can eliminate biases introduced by human experts and enable automation. These systems offer innovative solutions by providing more objective and consistent analysis, improving the accuracy of detection and identification. As technology advances, these automated systems will enhance efficiency and reliability in security and forensic applications by reducing human error and improving effectiveness.

## Anomaly activities detection from violent and crime scene videos

Anomaly detection using gait in crime scene videos involves the identification of abnormal walking patterns or body poses exhibited by individuals captured in video footage for criminal investigation and surveillance. As gait analysis is a valuable tool in identifying unique characteristics and deviations in how people walk and can determine human poses using pose estimation, it can provide important clues in solving crimes. Various deep learning techniques and algorithms are employed in the analysis of gait features. These may include computer vision methods such as optical flow, which tracks the movement of key points on a person's body, or deep learning models that can learn and recognize patterns in gait and body poses. In this section, several approaches for the detection and classification of anomaly activities in crime scene videos are discussed, and currently available crime-related video datasets are presented in Table 7.

The process of anomaly detection using gait in crime scene videos typically involves several steps. Research done by *Razak et al. (2022)* used CNN for the detection of house-breaking events based on human activities. They utilized CNN, and experiments were done

with several CNN architectures based on variant sizes, numbers of filters, and order of convolutional layers. In their study, the CNN model is trained to recognize four postures that happened in house-breaking activities, which are squatting, bending, squatting with heels up, squatting with heels down, and kneeling with heels. The dataset used for training contains 9558 images for each normal and anomaly class, and their best model achieved 97% accuracy. Another practical usage of gait is to detect intoxicated persons from gait data. *Li et al. (2023a)* proposed a CNN model that can differentiate intoxicated and sober individuals using gait information. The motion information from the accelerometer and gyrometer of the mobile phone is accumulated and transformed into Gramian Angular Field images and used as input to the model for intoxication detection.

Another work by *Boekhoudt et al. (2021)* applied encoder–decoder architecture called MPED-RNN (*Morais et al., 2019*) with HR-Crime dataset. The HR-Crime dataset contains 13 types of criminal activities and normal activity, shown in Table 7. The architecture of MPED-RNN consists of a combined encoder–decoder and RNN architecture to process sequential data with temporal information from input data. Moreover, rather than using sequences of images from the dataset, they used human poses extracted with YOLOv3 (*Farhadi & Redmon, 2018*) as input and evaluated with receiver operating characteristic (ROC) curve and its corresponding area under (AUROC), with which they attained performance score with 0.6030 AUROC. *Matei, Talavera & Aghaei (2022)* extended their research by adding data augmentation on input human poses. Their augmentation techniques include shifting skeletal joints coordinate and Synthetic Minority Oversampling Technique (SMOTE) (*Chawla et al., 2002*) to handle the imbalance dataset. They continued using MPED-RNN architecture for the classification of criminal activities.

By detecting anomalies in human gait and poses, law enforcement agencies can gain valuable insights into potential suspects or individuals involved in criminal activities. Anomalous patterns in gait and sequence of poses may indicate disguise, abnormal behavior, or attempts to conceal one's identity. Detecting these anomalies can aid in suspect identification, linking individuals to specific crime scenes, or providing additional evidence for criminal investigations. Furthermore, anomaly detection can be deployed in surveillance for security purposes. Anomaly detection in crime scene videos is an evolving field that benefits from advancements in computer vision, machine learning, and forensic science. By leveraging gait analysis techniques, law enforcement agencies can enhance their investigative capabilities and improve their ability to identify and apprehend suspects based on their unique walking patterns captured in crime scene videos.

## Gait analysis for surveillance

Surveillance systems, particularly those involving CCTV cameras, have increasingly relied on gait analysis for public safety and security. However, conventional gait analysis methods face significant challenges due to various covariate factors, such as changes in viewpoint, which can impact the accuracy of identification. To address this, *Kavitha & Gayathri (2022)* developed a deep CNN model using silhouette images extracted from input gait sequence and the model can identify the individual from any view angle even if the face is concealed. Moreover, gait information can be obtained from multiple sources and

smartphones are one of them and they provide rich information. *Sabir et al. (2019)* used gait data acquired from accelerometer sensors of smartphones to identify gender for safety and security purposes. One of the benefits of their approach is it requires little interaction with the individuals which is prominent for surveillance and public safety. They applied an RNN-based deep learning algorithm for gender identification and achieved high accuracy.

Another approach for processing sequential information like videos, RNN-based approaches is suitable, especially for videos and sequences of images attained from surveillance videos. *Topham et al. (2022)* integrated joint motion data of individuals from a sequence of gait cycle images for person identification using the LSTM model. Their input data can handle large sequences of data and, by applying a based approach they achieved 100% accuracy for person identification. Their approach is beneficial for public areas like airports and can overcome occlusion problems. Gait-based pedestrian identification is also a research area that covers public surveillance. The challenges lie in it is vast diversity of walking conditions and complex environment. *Wang & Yan (2023)* proposed for gait-based pedestrian identification using GRU module using gait energy images which represent spatiotemporal gait features focusing only on gait information and eliminating other unnecessary information and features. One of the innovative approaches is done by *Dandamudi et al. (2020)* for multiple object detection of activities, gender, weapons, fire, and sparkling wires on the roadside. The model is implemented with CNN and intended to be used for aerial surveillance using drones. Their model is not only able to handle multiple tasks but is also lightweight and implemented on IOT devices like Raspberry Pi and Arduino. The main intention of conducting this research is for public safety, especially to detect violence on women and, sexual and gender-based violence from pedestrian activities.

## Human pose estimation in forensic and criminal investigation

Human poses play a crucial evidence in forensic analysis for prosecuting crimes and can overturn wrongful convictions. One of the methods of analyzing the human poses is to generate an estimated weight and height from the human pose. Some of the basic and conventional forensic techniques are unreliable and generate false estimations in height and weight. To address this issue, *Thakkar & Farid (2021)* developed a 3D pose estimation method to estimate the height and weight of an individual from an image with high reliability. Moreover, in forensic science, analysis of dead bodies is involved to determine the cause of death. *Werukanjana et al. (2023)* developed dead body detection and pose estimation using YOLOv8 to assist investigators in Crime Scene Investigation. On the other hand, *Abruzzo et al. (2019)* proposed an approach for the detection of a person with a handgun from images and threat assessment based on the posture of that person. The model is developed using CNN and involves 2 stages. The first stage is to detect a person with a handgun and if the threat is detected, threat assessment is calculated based on the joint positions of the skeletal pose in the second stage.

## Gait verification for criminal investigation

Gait verification is a technique used in criminal investigations to identify and match individuals based on their unique walking patterns or gait characteristics. Gait can be

influenced by factors such as body structure, posture, movement patterns, and individual habits. By analyzing gait patterns captured in surveillance videos or other recordings, investigators can compare and verify the identity of individuals involved in criminal activities.

Gait verification involves extracting relevant features from gait sequences and employing various computational methods for analysis. These methods often utilize computer vision techniques, such as image processing and pattern recognition, to extract and compare gait features. Machine learning algorithms, such as deep neural networks (DNNs), can be trained on a database of known gait patterns and features to perform gait verification and identification. By comparing these extracted features with a database of known gait patterns, investigators can determine the likelihood of a match and use it as evidence in criminal investigations.

Gait-based verification system by *Iwama et al. (2012a)* utilized gait image sequences and CNN architecture to identify individuals and provide insights for forensic science. One of the purposes of developing the system is to allow non-specialists to analyze gait images. Their system accepts sequences of gait images and converts them to silhouette images. The generated human silhouette images are then processed into GEI, and they can be compared with the existing dataset for identification. The accuracy score for the identification of individuals is presented with probability. They also included a view transformation model, which transforms gait features from one view to another view. By combining multiple views, their system could overcome changes in view and clothing.

The use of gait verification in criminal investigations offers several advantages. It can provide additional evidence in cases where other identifying factors, such as facial features, are not clear or available. Moreover, gait analysis can be conducted from a distance or in low-resolution videos, making it useful in scenarios where direct observation or facial recognition may be challenging. Furthermore, gait verification can be applied in both real-time surveillance and forensic investigations (*Iwama et al., 2013*), helping to establish links between suspects and crime scenes.

## Limitations on applicability on practical scenarios

The implementation of gait analysis systems in real-world scenarios comes with several challenges. A major challenge lies in the complexity and variety of gait patterns, which can be influenced by numerous factors such as clothing, footwear, the surface on which a person is walking, and physical or emotional state. These variations can affect the accuracy of gait analysis systems, making it difficult to maintain consistent performance in real-world conditions. Other factors such as changes in lighting, camera angles, and complex backgrounds in surveillance videos could also complicate the accuracy and reliability of these systems.

Another significant challenge is the issue of data privacy and ethics. The use of gait analysis for constant monitoring and tracking of individuals raises privacy concerns. The standard to balance between security and the protection of individual rights is needed.

Additionally, the computational complexity involved in implementing gait analysis systems creates a hindrance to adoption. Processing large volumes of video data in real

time requires high computational resources, which can be costly and may not be applicable in resource-constrained environments.

Despite the challenges, gait analysis systems offer significant benefits for improving security and assisting with criminal investigations. As deep learning and technology continue to advance, many of these challenges are expected to be addressed. This will make gait recognition more practical and effective for use in real-world situations.

## COMPLEXITIES AND CHALLENGES

In criminal investigations and security monitoring, reviewing surveillance and crime scene footage is essential. Analyzing the human gait and behaviors depicted in the videos can offer insightful information and be used for crime prevention and detection. Lingering, unexpected movements, or carrying suspicious objects are just a few examples of unusual behavior that might be reported for additional inquiry. Videos may include important forensic evidence, like identification of weapons, clothing details, and license plate information. Investigations into crimes can benefit greatly from the extraction and enhancement of this evidence. Finding dangers or unusual events can be made easier by spotting anomalies in video data. Alerts for rapid action might be sent out in response to unusual crowd behavior, abandoned objects, or unauthorized access.

The performance of gait analysis has improved over time, and experts are continuing to enhance them. With the availability of gait datasets, several deep learning researchers have grown interested in developing deep learning-based gait analysis in several areas, such as healthcare and biometric authentication. The applicability of gait analysis using deep learning in criminal investigations involves several difficulties, and they are briefed in the following subsections.

### Developing deep learning models

A challenge in developing deep learning models is that they are often seen as "black boxes" and are difficult to understand and interpret. Although they can achieve high accuracy in identifying individuals, they can still be problematic in usability and applicability as evidence in forensic analysis. The ongoing research on explainable AI can be applied in this scenario, such as using visualization to understand the internal workings of deep learning models. Nonetheless, this can provide insights into how they are processing; the trade-off between accuracy and interpretability can happen. Moreover, implementing deep layers and complex models can still be hard to interpret.

One of the disadvantages of developing a deep learning model is its requirement for large data to train. With its complex nature and large amount of input data, choosing a suitable framework for deep learning architecture can be tedious. Although CNN-based framework can be chosen for image data and CNN-RNN combined framework for temporal information included data basically, other parameters such as the number of neural network layers, choosing the type of layer, learning rate, and so on, should be accounted and results to expensive training time and computational power. The complexity of the model lies in the complexity of the data and desired outcomes for the model.

## Lack of standard

Although gait-based criminal investigations produce results that are comparable, forensic gait analysis and gait recognition use different analysis techniques. While gait recognition seeks to construct person identification algorithms that may or may not use anatomical and physiological principles, forensic gait analysis draws on notions from clinical gait analysis. Most of the research on gait analysis is clinical and biomechanics-based and uses cutting-edge measurement techniques for accurate results in medical diagnosis and therapy. To demonstrate the method's usefulness and effectiveness in practice, data gathering for forensic methods should, nevertheless, try to mimic real-life events to acquire realistic results with quantified error rates.

Any forensic method's main objective is to offer justice-focused data that complies with the laws of the applicable legal system to be admitted as evidence in court. Particularly in disciplines involving pattern matching, such as gait analysis and recognition, inadequate analytic processes can result in biased data interpretation and cognitive bias. The forensic science approach balances objective interpretations with empirical data, yielding probabilistic results as opposed to unalterable facts. The degree of assurance and confidence in the "truth" differs across various forensic science disciplines. The use of invalidated procedures, peer review, and lack of validation can have an impact on whether guilt or innocence is proven in court (*Li et al., 2023b*).

## Lack of datasets

Availability of datasets and well-annotated datasets are vital for gait analysis research. To create precise and trustworthy algorithms and deep learning models for identifying people based on their walking patterns, gait analysis uses vast datasets. Gait analysis in criminal investigations faces a hindrance because there aren't enough acceptable datasets available. Videos involving crimes mostly come from surveillance, and the quality of videos is not enough for gait analysis to produce precise results. It also consists of recording video or motion-capture data of people moving while in different situations and environments, frequently with synchronized measurements of relevant ground truth data like identity and demographics (*Mangone et al., 2023*).

The development and generalizability of gait analysis techniques may be hampered by a lack of varied and consistent datasets. The enormous range of variation in human walking patterns, including differences in demographics, walking speeds, clothing kinds, and ambient conditions, may not be well reflected by the small number of these available datasets. As a result, deep learning models developed using tiny or biased datasets may struggle when used with populations or situations that are not well represented in the training data.

It is necessary to make efforts to gather, curate, and disseminate extensive and varied datasets to address this issue. Collaboration between academic institutions and other domains like law enforcement and security companies can aid in combining resources and knowledge to produce larger, more representative datasets. The dissemination and accessibility of gait datasets can be facilitated by open data initiatives and sharing platforms, allowing researchers to create and evaluate reliable gait analysis techniques (*Vedwal, 2023*).

## Presentation of likelihood calculation

In gait analysis, the process of determining and quantifying the likelihood or confidence that a specific gait-related event or characteristic will occur in a video sequence is known as the presentation of likelihood computation in video footage. To assess whether particular gait characteristics are present in the video clip, visual features and patterns are taken from it and analyzed as part of the likelihood calculation. Stride length, step breadth, cadence, joint angles, and other pertinent factors that contribute to the features of a person's gait may be included in likelihood calculation.

There are many ways to present the likelihood calculation. Assigning a score or numerical number to reflect the probability of seeing a particular gait parameter is one frequent method. This score can be between 0 and 1, with 0 signifying a low frequency of occurrence or absence and 1 signifying a high likelihood of occurrence or presence. Graphical representations like histograms, line plots, or bar charts can be used to visualize the presentation of likelihood computation. These graphics offer a simple method to comprehend how likelihood ratings are distributed among various gait metrics or people (*Ye et al., 2023*). Additionally, the spatial and temporal fluctuations in likelihood scores can be represented using color-coded heat maps. With the help of these heat maps, which are superimposed over the video frames and provide areas of higher likelihood of a distinct hue or intensity, gait occurrences or irregularities can be visually interpreted.

It is crucial to remember that when the likelihood calculation is presented, the results' significance or dependability should be assessed using the relevant statistical metrics or thresholds. By using these metrics, one can better understand the likelihood scores and make decisions on the further processes like forensic analysis. For instance forensic experts need to provide a clear explanation of the methodology used for calculation, including the parameters and variables considered while submitting as evidence in court.

Despite its potential, the presentation of likelihood calculations in gait analysis can raise concerns in a legal setting. One significant concern is the need for a robust scientific foundation. The methodology behind likelihood calculations should be based on well-established principles and validated techniques. With the absence of a consensus standard or protocol, presenting likelihood from gait analysis as evidence raises concerns about its reliability and admissibility in court (*Macoveciuc, Rando & Borrion, 2019*).

Overall, the gait analysis presentation of likelihood calculation to provide a clear and understandable representation of the confidence or probability of gait metrics or events detected in video footage is crucial, permitting subsequent study and decision-making processes in forensic or criminal investigation (*Khan, Farid & Grzegorzek, 2023*).

## Privacy and data protection

As gait can identify a human individual and is regarded as a biometric, processing and handling human gait data raises concerns about privacy and data protection. Maintaining the protection of gait information acquired from surveillance videos requires robust privacy and data protection measures. This also includes processing video data and images that contain human gait information to handle with a secure way when processing the data for training deep learning models. Recent of incidents of biometric data theft have raised

privacy concern as gait information comprises of an individual identity and biometric information.

Furthermore, without clear guidelines and regulations, the personnel handling the data may use gait analysis for malicious purposes and this could lead to a violation of individuals' privacy. By addressing these challenges by implementing robust privacy and data protection measures, and handle with accountability and secure way to prevent misuse, gait analysis can be used in a responsible and ethical manner for criminal investigation.

# FUTURE DIRECTIONS AND PROSPECTS

Applying deep learning has shown significant transformations in multiple areas and applying it in criminal investigation has offered multiple benefits and effectiveness. Though it has proven some improvements, several territories can still be explored and enhanced, particularly in multimodal inputs, multitask learning, explainable AI (XAI), and federated learning.

## Multimodal inputs

Conventionally, gait analysis for criminal investigation uses video footage or a sequence of images acquired from surveillance. Rather than depending on single data input, other modalities such as audio and sensor data can be integrated to enhance the accuracy and robustness of gait analysis. A multimodal approach can provide a more comprehensive understanding of gait patterns and increase the chances of identifying suspects in criminal investigations.

## Multitask learning

One of the abilities of the deep learning model is multitask learning in which a single model is trained for multitasks simultaneously and separate results are generated for each task (*Standley et al., 2020*). With multitask learning, the gait analysis model can be trained with the same dataset for suspect identification and criminal activity detection tasks. By training for similar or related tasks, the model can learn common features and reduce overfitting in model training by sharing parameters and representations. For instance, anomaly action detection and criminal activity recognition are two similar tasks and they can be trained in the same model with multitask learning.

## Explainable AI

The need for transparency and interpretability is essential when gait analysis results are admitted to the court as evidence. Explainable AI (XAI) ensures the reasoning behind the deep learning models' predictions is clear and precise (*Nutter, 2019*). XAI will play a crucial part regarding legal accountability and admissibility as evidence in court proceedings. By providing insights into the decision-making process and algorithms of deep learning models, XAI can establish the reliability and validity of the results in criminal investigations.

## Federated learning

Biometric data like gait information are sensitive data and, while processing and presenting these data raises privacy concerns, especially when large amounts of gait data are used

for model training from multiple sources. To address this, federated learning can offer a promising solution by allowing the model to be trained without seeing or touching the data (*Martineau, 2022*). In criminal investigations, data from several law enforcement agencies can collaborate on model training for gait recognition without compromising the privacy of the data or sharing sensitive data. Each agency can train a gait analysis model and share the model updates with other agencies by federated learning.

## CONCLUSION

In conclusion, gait analysis has emerged as a valuable tool in criminal investigations, offering a unique and non-invasive nature for identifying individuals and admissible evidence in court proceedings and forensic analysis. This article has provided a comprehensive review of the current state of gait analysis for criminal investigations, highlighting the potential benefits and challenges of using deep learning techniques and available datasets for gait recognition and criminal actions. According to the literature review, the use of deep learning techniques has significantly improved the accuracy and efficiency of gait analysis, enabling the extraction of relevant features from the data, like surveillance videos, and the development of robust models for gait recognition and criminal action detection. However, challenges remain to be addressed, such as inhabiting factors in gait regarding surveillance, developing explainable AI models for gait analysis, and handling gait biometric data with ethics and data protection. Future perspectives should focus on improving the accuracy and reliability of gait analysis, enhancing the effectiveness of criminal investigations. Addressing these challenges will be crucial for the continued development of deep learning-based gait analysis in criminal investigations. As technology continues to evolve, gait analysis will continue to play an increasingly important role in criminal investigations.

### Funding

This research project was supported by Scholarships for Ph.D. Student from Mahidol University. There was no additional external funding received for this study. The funders had no role in study design, data collection and analysis, decision to publish, or preparation of the manuscript.

### Grant Disclosures

The following grant information was disclosed by the authors:
Scholarships for Ph.D. Student from Mahidol University.

### Competing Interests

The authors declare there are no competing interests.

### Author Contributions

- Sai Thu Ya Aung conceived and designed the experiments, performed the experiments, analyzed the data, performed the computation work, prepared figures and/or tables, authored or reviewed drafts of the article, and approved the final draft.

- Worapan Kusakunniran conceived and designed the experiments, performed the experiments, analyzed the data, performed the computation work, prepared figures and/or tables, authored or reviewed drafts of the article, and approved the final draft.

## Data Availability

This is a literature review.

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
