# Peer review of "A comprehensive review of gait analysis using deep learning approaches in criminal investigation"

_PeerJ Computer Science, doi:10.7717/peerj-cs.2456_

## Round 0.1 · original submission · Major Revisions

Dear authors,

You are advised to critically respond to all comments point by point when preparing a new version of the manuscript and while preparing for the rebuttal letter. Please address all the comments/suggestions provided by the reviewers.

Kind regards,
PCoelho

Reviewer 1 ·

Basic reporting

Literary work needs revision highlighting motivation and need for conducting the review.
Structuring and English needs revisit

Experimental design

Methodology is not described as required.

Validity of the findings

The article provides a comprehensive review about existing literature, however, my concerns are:

1. The motivation to carry out the review is missing. How the conducted review add a piece of information to existing literature.
2. Everly progress should be added in the review to highlight the significance of literary work.
3. A graphical flow to should the adopted methodology is required.
4. The article screening involves limited keywords, how bias with other studies is handled.
5. The most efficient DL model in gait analysis and criminal investigation or about the same variants has to be recorded to aid the researchers working in the same area.
6. A meta analysis or cumulative result about DL models determining strength and weakness, computational constraints needs to be elaborated.
7. The whole structure needs to be précised and outcome oriented rather than just describing each study.
8. Concise conclusion and importance with more focused future perspectives is needed.

·

Basic reporting

The paper is well-written, with professional language that is clear and engaging.

Regarding the Review Methodology section, it could benefit from further details on how the relevance of the articles was determined, how much papers were selected before the screening phases, how much paper were discarded in each screening phase and the number of selected papers, improving the transparency and rigor of the review process.

The paper should provide more concrete examples of successful applications of gait analysis in criminal investigations, to better illustrate its practical utility.

Experimental design

The transition between sections should be smoother to improve readability, especially during the chapters DEEP LEARNING BASED GAIT ANALYSIS APPROACHES and RECENT RESEARCHES ON GAIT ANALYSIS FOR CRIMINAL INVESTIGATION. Furthermore, the review could be improved by discussing emerging trends and future directions in gait recognition, such as multimodal data integration or advances in 3D gait recognition , which are critical areas for future research and development.

Validity of the findings

Emphasizing the practical applications and challenges of implementing gait recognition systems in real-world scenarios would provide a more balanced perspective. Discussing these aspects would not only highlight the current capabilities and limitations of gait recognition technology, but also address potential concerns and barriers to implementation in various practical settings. By addressing these minor revisions, the paper can further increase its value and impact, offering comprehensive insights and practical guidance for both researchers and practitioners in the field of gait analysis.

---

## Round 0.2 · Minor Revisions

Dear authors,
Thanks a lot for your efforts to improve the manuscript.
Nevertheless, some minor concerns from R1 are still remaining that need to be addressed.

Like before, you are advised to critically respond to the remaining comments point by point when preparing a new version of the manuscript and while preparing for the rebuttal letter.

Kind regards,
PCoelho

Reviewer 1 ·

Basic reporting

The authors have addressed all the concerns raised.

The paper can be accepted for inclusion in journal.

However, detailed information is required for the structure of various layers in deep learning techniques in figures (Figure 4-6) that is, name of layer with its structural parameters that would be more effective for the readers.

Experimental design

Review Paper

Validity of the findings

Improvements are included as suggested

Additional comments

Nil

·

Basic reporting

it is improved

Experimental design

now is correct

Validity of the findings

sufficient

Additional comments

none

---

## Round 0.3 · accepted · Accept

Dear authors, we are pleased to verify that you meet the reviewer's valuable feedback to improve your research.

Thank you for considering PeerJ Computer Science and submitting your work.